# Hedgehog signaling patterns the oral-aboral axis of the mandibular arch

**Jingyue Xu[1], Han Liu[1], Yu Lan[1,2,3,4], Mike Adam[1], David E Clouthier[5], Steven Potter[1,3], Rulang Jiang[1,2,3,4]\***

[1]Division of Developmental Biology, Cincinnati Children's Hospital Medical Center, Cincinnati, United States; [2]Division of Plastic Surgery, Cincinnati Children's Hospital Medical Center, Cincinnati, United States; [3]Department of Pediatrics, University of Cincinnati College of Medicine, Cincinnati, United States; [4]Shriners Hospitals for Children – Cincinnati, Cincinnati, United States; [5]Department of Craniofacial Biology, School of Dental Medicine, Anschutz Medical Campus, University of Colorado, Aurora, United States

**Abstract** Development of vertebrate jaws involves patterning neural crest-derived mesenchyme cells into distinct subpopulations along the proximal-distal and oral-aboral axes. Although the molecular mechanisms patterning the proximal-distal axis have been well studied, little is known regarding the mechanisms patterning the oral-aboral axis. Using unbiased single-cell RNA-seq analysis followed by in situ analysis of gene expression profiles, we show that Shh and Bmp4 signaling pathways are activated in a complementary pattern along the oral-aboral axis in mouse embryonic mandibular arch. Tissue-specific inactivation of hedgehog signaling in neural crest-derived mandibular mesenchyme led to expansion of BMP signaling activity to throughout the oral-aboral axis of the distal mandibular arch and subsequently duplication of dentary bone in the oral side of the mandible at the expense of tongue formation. Further studies indicate that hedgehog signaling acts through the Foxf1/2 transcription factors to specify the oral fate and pattern the oral-aboral axis of the mandibular mesenchyme.

DOI: https://doi.org/10.7554/eLife.40315.001

**\*For correspondence:**
rulang.jiang@cchmc.org

**Competing interests:** The authors declare that no competing interests exist.

## Introduction

Formation of the jaws conferred a huge evolutionary advantage to the early vertebrates such that more than 99% of living vertebrates today have biting jaws as a primary feeding apparatus (*Brazeau and Friedman, 2015*; *Janvier, 1996*; *Miyashita, 2016*). Central in the formation and evolution of the vertebrate jaws is the embryonic first pharyngeal arch, the rostral-most pharyngeal arch recognized in the embryo (*Cerny et al., 2004*; *Kuratani, 2004*; *Kuratani, 2012*; *Miyashita, 2016*). Whereas all of the pharyngeal arches form by cranial neural crest cells that populate between the embryonic foregut endoderm and ventral surface ectoderm, and although neural crest cells populating each of the other pharyngeal arches are specified by a combination of colinearly expressed *Hox* genes, the neural crest cells populating the first arch are *Hox*-negative and their fates are primarily regulated by signaling molecules expressed in the arch epithelium (*Minoux and Rijli, 2010*; *Miyashita, 2016*; *Trainor and Krumlauf, 2001*). As has been demonstrated by cell lineage studies in both axolotl and chick embryos, neural crest cells in the dorsal portion of the first pharyngeal arch contribute to trabecular bones of the neural cranium whereas neural crest cells in the ventral portion of the first arch, referred to as the mandibular arch, give rise to skeletal elements of both the upper and lower jaws (*Cerny et al., 2004*; *Kuratani et al., 2013*). In addition to giving rise to the skeletal structures of the jaws, which form in the aboral regions of the maxillary and mandibular prominences (*Cobourne and Sharpe, 2003*), the neural crest-derived ectomesenchyme cells interact with the oral

epithelium to form teeth as well as to organize tongue formation at the oral side of the mandible (*Cobourne and Sharpe, 2003*; *Lumsden, 1988*; *Mina and Kollar, 1987*; *Parada and Chai, 2015*). Whereas previous studies have uncovered detailed molecular mechanisms patterning the rostral-caudal and proximal-distal axes of the jaw skeleton (*Medeiros and Crump, 2012*; *Minoux and Rijli, 2010*), the molecular mechanism patterning the oral-aboral axis of the mandibular arch mesenchyme is not well understood.

Upon arriving in the mandibular arch, the neural crest cells encounter a wealth of epithelial signals, including Fgf8 and Bmp4 expressed by the facial ectoderm as well as Shh expressed by the pharyngeal endoderm (*Brito et al., 2006*; *Haworth et al., 2004*; *Haworth et al., 2007*; *Jeong et al., 2004*; *Shigetani et al., 2000*; *Tucker et al., 1998*; *Tucker et al., 1999*). Both Fgf8 and Shh signaling are crucial for the survival and proliferation of the neural crest cells colonizing the mandibular arch (*Brito et al., 2006*; *Creuzet et al., 2004*; *Jeong et al., 2004*; *Trumpp et al., 1999*). Fgf8 and Bmp4 act to pattern the proximal-distal axis of the mandibular arch by activating expression of distinct homeodomain-containing transcription factors in the neural crest-derived mandibular mesenchyme, including Barx1 expression in the proximal region and Msx1 and Msx2 in the distal region (*Barlow et al., 1999*; *Ferguson et al., 2000*; *Tucker et al., 1998*). In addition, it has been shown that restricted expression of the Lim-homeobox domain genes *Lhx6* and *Lhx8* (previously called *Lhx7*) in the rostral region of the mandibular mesenchyme depends on Fgf8 signaling from the mandibular ectoderm (*Cobourne and Sharpe, 2003*; *Grigoriou et al., 1998*; *Tucker et al., 1999*). Since *Lhx8* mRNA expression was found restricted in the rostral region of the mandibular arch mesenchyme on frontal sections, the authors interpreted the rostral side of the mandibular arch as the oral side and suggested that Fgf8 signaling might be important in patterning the oral-aboral axis of the mandible (*Cobourne and Sharpe, 2003*; *Grigoriou et al., 1998*; *Tucker et al., 1999*). However, tissue-specific inactivation of *Fgf8* in the early mandibular arch epithelium in the mouse embryos caused complete loss of proximal mandibular structures (*Trumpp et al., 1999*), which proved that Fgf8 signaling is essential for proximal mandibular development but whether Fgf8 signaling is required for patterning the oral-aboral axis remains unresolved.

Recent development of the single cell RNA-seq (scRNA-seq) technology allows simultaneous profiling of the transcriptomes of thousands of individual cells from an organ or tissue in a single experiment and is revolutionizing many areas of biology and disease research (*Klein et al., 2015*; *Macosko et al., 2015*; *Park et al., 2018*; *Rosenberg et al., 2018*; *Tirosh et al., 2016*; *Venteicher et al., 2017*; *Zheng et al., 2017*). In addition to identifying novel cells as well as uncovering new marker genes for previously defined cell types within both healthy and diseased tissues (*Klein et al., 2015*; *Park et al., 2018*; *Rosenberg et al., 2018*; *Tirosh et al., 2016*; *Venteicher et al., 2017*; *Zheng et al., 2017*), scRNA-seq has enabled effective generation of high resolution molecular atlases of the gene expression programs that drive tissue and organ morphogenesis (*Adam et al., 2017*; *Park et al., 2018*; *Rosenberg et al., 2018*). In this study, we performed scRNA-seq analysis of E10.5 mouse mandibular arch. Unbiased clustering analysis of more than 10,000 cells from the E10.5 mandibular arch confirms the compartmentalized expression of major signaling molecules, including Bmp4, Fgf8, Shh, Wnt6, in the mandibular epithelium and spatially-restricted target gene expression in the mandibular arch mesenchyme. Whereas previous reports of whole mount in situ hybridization data showed restricted expression of several Bmp and Shh target genes in the distal mandibular mesenchyme at this developmental stage (*Jeong et al., 2004*; *Liu et al., 2005*), our scRNA-seq data revealed that these two pathways are activated in complementary subdomains in the distal mandibular arch mesenchyme. Subsequent in situ hybridization analysis validated these findings and revealed that expression of Shh and Bmp4 target genes are patterned along the oral-aboral axis. Furthermore, we show that tissue-specific inactivation of *Smo*, which encodes an obligatory and cell-autonomous transducer of hedgehog signaling (*Briscoe and Vincent, 2013*; *Jeong et al., 2004*; *Zhang et al., 2001*), in the distal mandibular arch mesenchyme led to duplication of the dentary bone in the oral side of the mandibular mesenchyme at the expense of tongue formation. In the absence of *Smo* function, BMP signaling is activated throughout the oral-aboral axis of the distal mandibular arch mesenchyme. Moreover, we show that tissue-specific inactivation of *Foxf1* and *Foxf2* in the cranial neural crest cells also led to agenesis of the oral tongue structure and ectopic bone formation in the oral side of the mandible. These results uncover a previously unknown molecular mechanism involving antagonistic actions of hedgehog and BMP signaling in patterning the oral-aboral axis of the mandible.

## Results

### Single-cell RNA-seq analysis of the E10.5 mouse mandibular arch

To gain better understanding of the gene expression programs driving mandibular morphogenesis, we performed scRNA-seq analysis of E10.5 mouse mandibular arch using the 10x Genomics's Chromium Single Cell system (see Materials and methods). After filtering at both cell and gene levels using highly stringent criteria (see Materials and methods), we obtained high quality transcriptome profiles of 10,586 cells, which expressed 17,074 unique genes, with the median number of detected genes and transcripts at 2624 and 8824, respectively. The unsupervised clustering of the scRNA-seq data generated the cell groupings shown in a two-dimensional t-distributed Stochastic Neighbor Embedding (tSNE) plot in *Figure 1A*. All major cell types were identified and validated by the expression of known cell type selective markers. The transcriptome signature gene list for the cell clusters are shown in *Supplementary file 1* and the gene expression profiles summarized in a heatmap in *Figure 1—figure supplement 1*. As expected, the majority of the cells were of neural crest origin, accounting for over 9000 of the 10,586 cells and are distributed in two subgroups, NC1 and NC2, representing cells in the proximal (expressing *Barx1*) and distal (expressing *Hand2*) regions of the E10.5 mandibular arch, respectively. About 8% of the cells were epithelial cells, which are marked by expression of *Epcam* (830 cells, including both endoderm and ectoderm cells). The endothelial cells (234 cells, marked by *Pecam1* expression) and other mesoderm-derived cells (249 cells) were also grouped according to their cell type identity. The NC3 cluster, consisting of 484 cells, is distinguished from the other clusters primarily by overall lower levels of transcripts and dramatically lower number of genes detected per cell (*Figure 1—figure supplement 2* and *Figure 1—figure supplement 3*, and *Supplementary file 2*). In addition, the NC3 cluster cells showed very low levels or absence of several ubiquitously expressed, nuclear localized long noncoding RNAs, including *Malat1*, *Xist*, *Meg3,* and *Kcnq1ot1* (*Supplementary file 2*, *Figure 1—figure supplement 4*). These data suggest that the NC3 cluster resulted, most likely, from incomplete lysis of a subset of single cells.

We performed iterative clustering analysis of the neural crest cell populations from NC1 and NC2 clusters shown in *Figure 1A*, which further grouped the neural crest-derived mandibular mesenchyme cells into six subgroups. Over/under clustering was verified via gene expression heatmaps. A 3D tSNE projection of these neural crest cell subgroups is shown in *Figure 1—video 1* and a 2D tSNE projection is shown in *Figure 1B*. A list of highly differentially expressed marker genes of each of the six clusters is provided in *Supplementary file 3*. Subgroups 0 and 1 share high levels of expression of *Hand2*, a marker of the distal domain of the mandibular arch mesenchyme at this developmental stage, whereas Subgroup 2 lacks *Hand2* expression and instead expresses the proximal domain marker genes *Emx2* and *Pou3f3* at high levels (*Figure 1C*). Subgroups 3 and 4 appear to distribute along the rostral-caudal axis, with Subgroup 3 expressing high levels of the caudal domain marker gene *Gsc* and with Subgroup 4 expressing high levels of the rostral domain marker genes *Lhx8* and *Lhx6* (*Figure 1B,C*). Subgroup 5 cells are more scattered than the other subgroups in the tSNE plot but are uniquely marked by high levels of expression of *Notch2* (*Figure 1B,C*, *Supplementary file 3*).

Whereas it is remarkable that the iterative clustering analysis of the scRNA-seq data grouped the mandibular neural crest mesenchyme cells into subgroups that occupy distinct spaces along the proximal-distal and rostral-caudal axes, most of the marker genes are not exclusively expressed in only one subgroup and there was no sharp boundary between the adjacent subgroups (*Figure 1B, C*, *Supplementary file 3*). Nevertheless, the cellular distributions in the tSNE plot correlated well with their spatial patterns of gene expression along the proximal-distal and rostral-caudal axes of the mandibular arch in the intact embryo (*Figure 2*). Moreover, although both Subgroups 0 and 1 consist of neural crest cells from the distal region of the mandibular arch mesenchyme marked by high *Hand2* expression, they are distinguished from each other by significant differential expression of multiple genes (*Figure 1C*, *Supplementary file 3*). Strikingly, several Hedgehog signaling target genes, including *Ptch1*, *Foxf1*, and *Foxf2*, were among the top marker genes for Subgroup 0 whereas several BMP signaling target genes, including *Msx1*, *Msx2*, and *Bambi* were among the top markers for Subgroup 1. Visualization of the 3D tSNE projection of all six neural crest cell subgroups suggest that Subgroups 0 and 1 are distributed along a distinct axis from Subgroups 2 and 3. Furthermore, gene ontology analyses showed 'skeletal system development', 'ossification', and

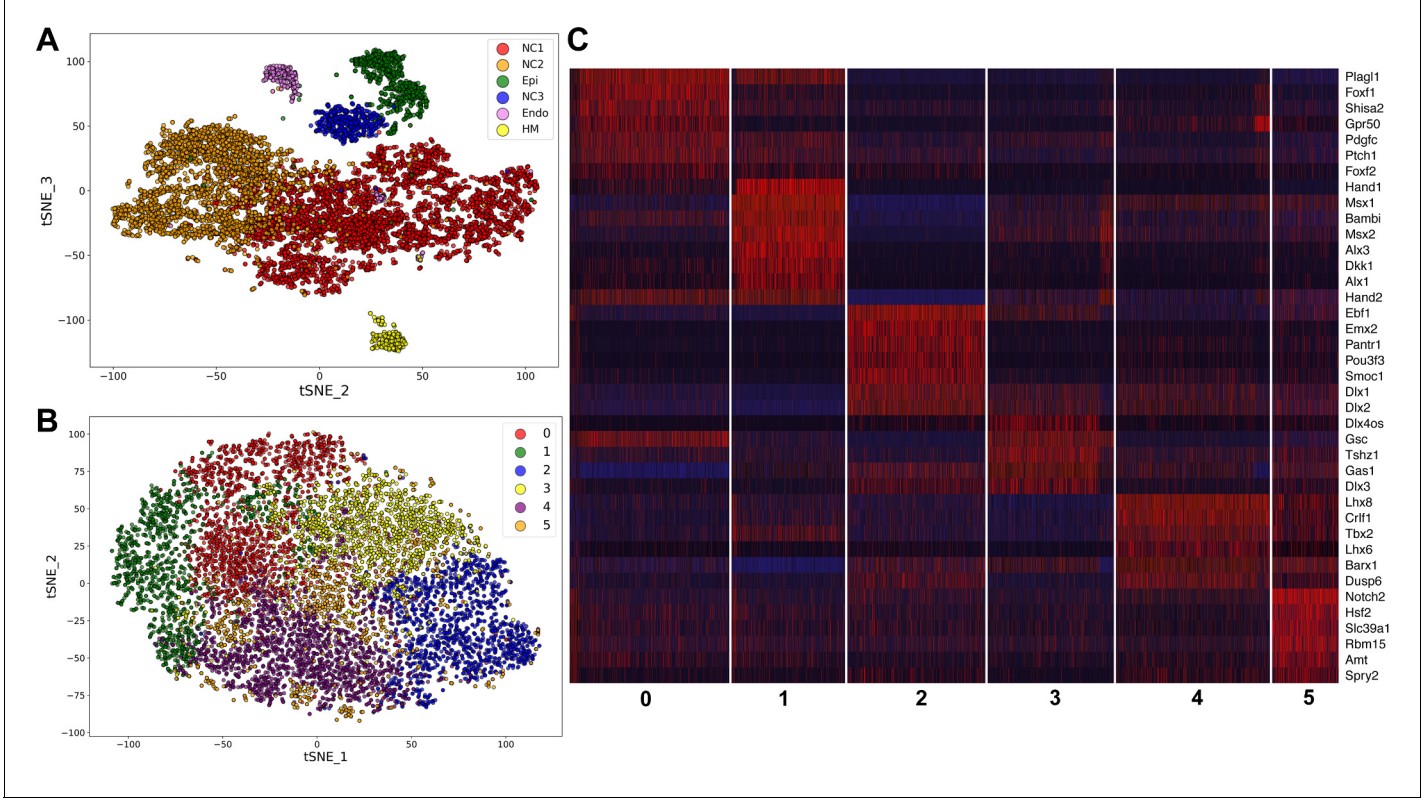

**Figure 1.** Analysis of scRNA-seq data of E10.5 mouse mandibular arch. (**A**) tSNE plot of all cells with high quality RNA-seq results shows clustering of the major cell types. (**B**) Unsupervised clustering of the neural crest-derived mandibular mesenchyme cells shows six identifiable subgroups. (**C**) Heat map showing distribution of cells in the six subgroups shown in B, with selected marker genes exhibiting more than 1.3-fold difference in expression levels across all six subgroups. The cluster numbers are marked at the bottom of Panel C and correspond to the cluster numbers in Panel B. The expression level for each marker gene in the individual cells is reflected by color of the vertical bars, with bright blue being lowest and bright red being highest.

DOI: https://doi.org/10.7554/eLife.40315.002

The following video and figure supplements are available for figure 1:

**Figure supplement 1.** Heatmap showing transcriptome profiles of the major cell types in the E10.5 mouse mandibular arch, with select marker genes exhibiting more than 1.5-fold difference in expression levels across all clusters.

DOI: https://doi.org/10.7554/eLife.40315.003

**Figure supplement 2.** Dot plot showing the number of unique genes whose expression was detected in the single cells in the six different clusters.

DOI: https://doi.org/10.7554/eLife.40315.004

**Figure supplement 3.** Dot plot showing the number of transcripts detected in the single cells in the six different clusters.

DOI: https://doi.org/10.7554/eLife.40315.005

**Figure supplement 4.** Comparison of the levels of transcripts for selected marker genes in tSNE maps of the E10.5 mandibular arch scRNA-seq data.

DOI: https://doi.org/10.7554/eLife.40315.006

**Figure supplement 5.** Distribution of the principal components by JackStraw plot.

DOI: https://doi.org/10.7554/eLife.40315.007

**Figure supplement 6.** Analysis of the top eight principal components identified in the scRNA-seq data.

DOI: https://doi.org/10.7554/eLife.40315.008

**Figure supplement 7.** Developmental trajectories of all of the neural crest cells derived from analysis of the E10.5 mandibular arch scRNA-seq data using Monocle-2.

DOI: https://doi.org/10.7554/eLife.40315.009

**Figure supplement 8.** Distribution of the five cellular states from the developmental trajectory analysis of the neural crest cells in the E10.5 mandibular arch on tSNE plot of the six subgroups derived from iterative clustering.

DOI: https://doi.org/10.7554/eLife.40315.010

**Figure 1—video 1.** A .mov file showing three dimensional tSNE projection of the six subgroups of neural crest derived mandibular arch mesenchyme cells.

DOI: https://doi.org/10.7554/eLife.40315.011

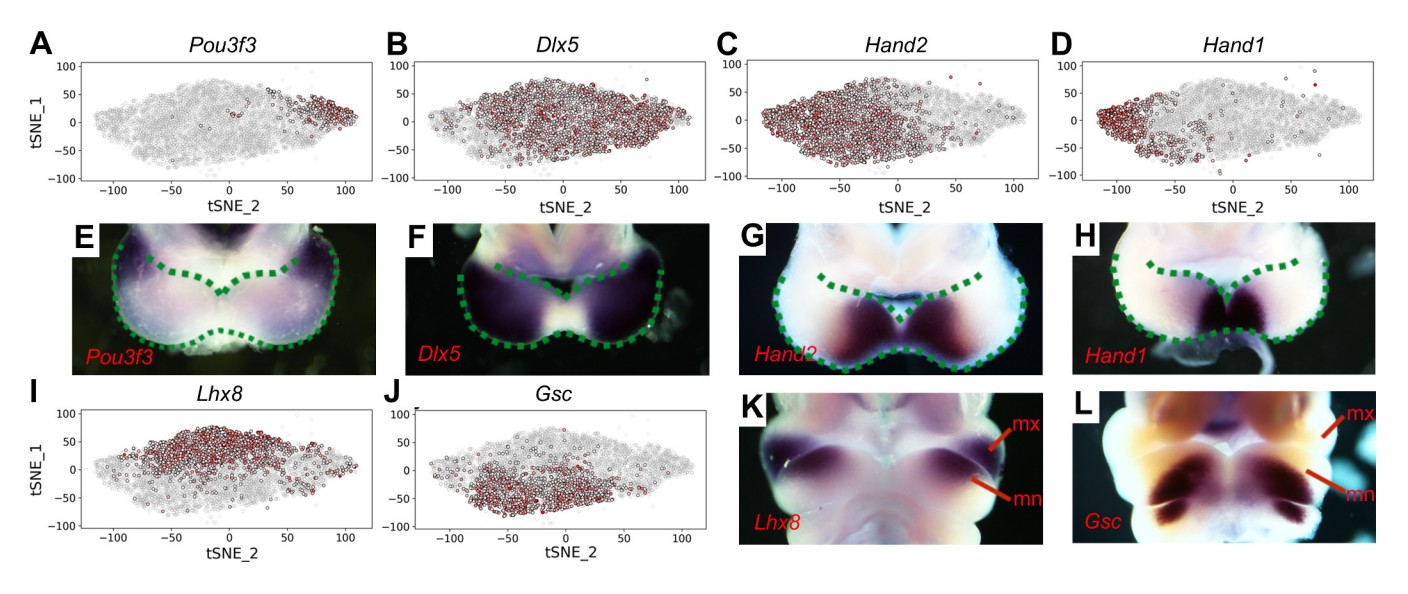

**Figure 2.** Distribution of E10.5 mandibular arch cells expressing select marker genes in the neural crest compartments in tSNE maps. (A–D) Cells expressing *Pou3f3* (A), *Dlx5* (B), *Hand2* (C), and *Hand1* (D) in the neural crest cell population, detected by scRNA-seq, are shown in red color in tSNE plot, with the brightness of the red color corresponding to levels of transcripts detected. (E–H) Rostral views of the mandibular arches of E10.5 mouse embryos showing patterns of expression of *Pou3f3* (E), *Dlx5* (F), *Hand2* (G), and *Hand1* (H) mRNAs detected by whole mount in situ hybridization. Green dots mark the oropharyngeal and aboral sides of the mandibular arches, with the distal caps of the paired mandibular arches meeting at the midline. (I, J) Cells expressing *Lhx8* (I), and *Gsc* (J) in the neural crest cell population, detected by scRNA-seq, are shown in red color in tSNE plot. (K, L) Frontal views of the mandibular arches of E10.5 mouse embryos showing patterns of expression of *Lhx8* (K) and *Gsc* (L) mRNAs detected by whole mount in situ hybridization. mn, mandibular process; mx, maxillary process.
DOI: https://doi.org/10.7554/eLife.40315.012

'osteoblastic differentiation' among the enriched biological processes for both Subgroups 0 and 1 (*Supplementary file 4* and *5*). These results suggest that the molecular pathways patterning Subgroups 0 and 1 maybe responsible for patterning the oral-aboral axis of the mandibular arch.

In addition to clustering analyses and visualization of the scRNA-seq data in 2D and 3D tSNE projections, we also analyzed significant principal components determined by JackStraw plot but the principal component analysis did not clearly distribute the neural crest cells along the major spatial axes (*Figure 1—figure supplement 5* and *Figure 1—figure supplement 6*). We also analyzed the developmental trajectories of all of the neural crest cells using Monocle-2 (*Qiu et al., 2017*), which distributed the mandibular arch mesenchyme into five different states (*Figure 1—figure supplement 7*). However, GO analysis showed that only States 3 and 4 showed enrichment for cell type differentiation, with State 3 significantly enriched for 'osteoblast differentiation' (*Supplementary file 6*) and State 4 enriched for 'neuron differentiation' (*Supplementary file 7*). State 3 contains only a small subset of the mandibular arch mesenchyme cells that are widely scattered on the tSNE plot (*Figure 1—figure supplement 8*). Altogether, the unsupervised iterative clustering analysis of the scRNA-seq data with 3D tSNE projections provided the highest resolution to distribute the mandibular arch single cells into subgroups along three major spatial axes.

To verify whether the Hedgehog and BMP signaling pathways are activated in a complementary pattern along the oral-aboral axis in the mandibular arch mesenchyme in vivo, we performed in situ hybridization and immunofluorescent staining analysis to directly compare the patterns of gene expression along the oral-aboral axis of the E10.5 mandibular arch. Previous studies of gene expression patterns in the developing craniofacial tissues in mouse or chick embryos often presented the data in frontal views of the embryonic facial prominences, which provide clear visualization of gene expression patterns along the proximal-distal and rostral-caudal axes of the mandibular arch (*Figure 3A–D*) but have rarely directly compared patterns of gene expression along the oral-aboral axis. For examining whole mount in situ hybridization data, we found that removing the frontonasal and maxillary regions and looking directly down at the mandibular arches of E10.5 mouse embryos,

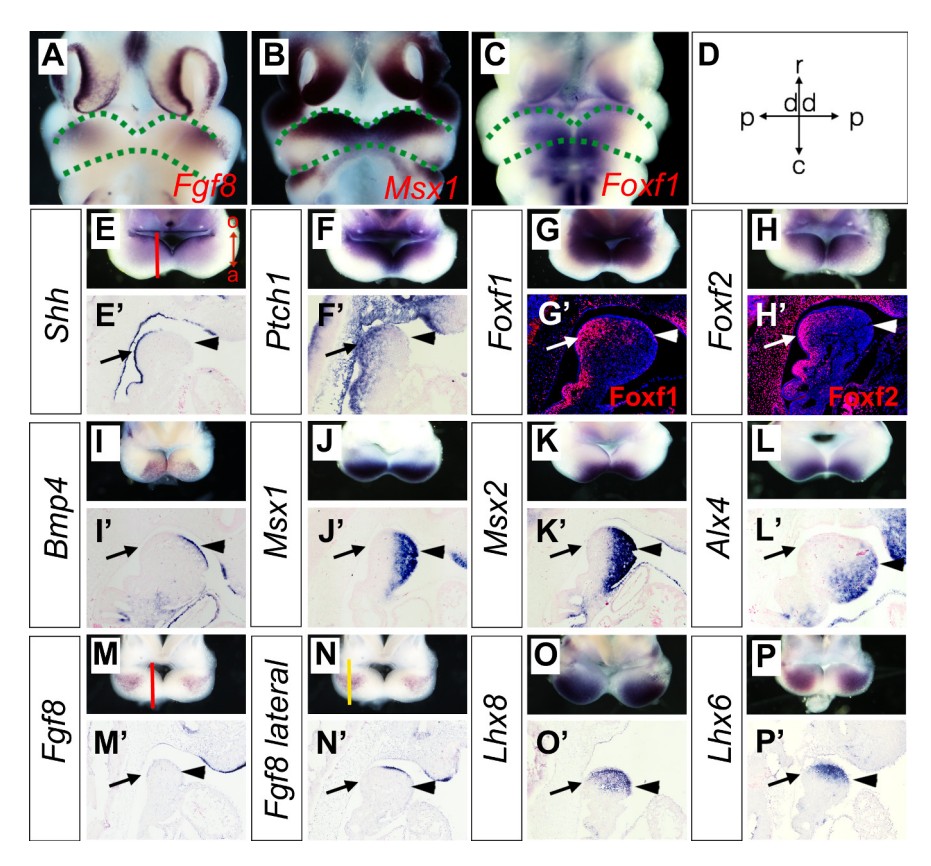

**Figure 3.** Expression of *Shh, Bmp4, Fgf8,* and their target genes in the developing mandibular arch. (A–C) Frontal views of E10.5 embryonic mouse heads after whole mount in situ hybridization detecting *Fgf8* (A), *Msx1* (B) and *Foxf1* (C). Green dashed lines mark the rostral and caudal sides of the embryonic mandibular arch. (D) A schematic indication of the proximal-distal and rostral-caudal axis of developing mandibular arch in the frontal view. (E–P) Rostral views of the mandibular arches in E10.5 mouse embryos showing patterns of expression of *Shh* (E), *Ptch1* (F), *Foxf1* (G), *Foxf2* (H), *Bmp4* (I), *Msx1* (J), *Msx2* (K), *Alx4* (L), *Fgf8* (M, N), *Lhx8* (O), and *Lhx6* (P) mRNAs. The red vertical line in Panel E shows approximate position of sagittal sections selected for in situ hybridization detection on sections. The orientation of the oral (o) and aboral (a) axis is also indicated in Panel E. Note the complementary pattern of Foxf1 and Msx1 expression along the oral-aboral axis in G and K that could not be distinguished on the frontal views in B and C. (E'–P') Sagittal sections showing expression of *Shh* (E'), *Ptch1* (F'), *Bmp4* (I'), *Msx1* (J'), *Msx2* (K'), *Alx4* (L'), *Fgf8* (M', N'), *Lhx8* (O'), *Lhx6* (P') mRNAs, and Foxf1 (G') and Foxf2 (H') proteins in the distal mandibular arch in E10.5 mouse embryos. Protein immunofluorescence is shown in red color in G' and H', whereas mRNA signals in all other panels are shown in purple/blue color. Arrow points to oropharyngeal endoderm, and arrowhead points to aboral ectoderm. The red and yellow lines in (M) and (N), respectively, indicate the different positions corresponding to the sections in (M') and (N').

DOI: https://doi.org/10.7554/eLife.40315.013

but not the frontal views of the embryonic mandible, provided clear visualization of the patterns of gene expression along the oral-aboral axis of the mandibular arch (*Figure 3A–P*). We further validated the patterns of gene expression along the oral-aboral axis by section in situ hybridization assays. Consistent with previous reports from studies of chick embryos (*Brito et al., 2008*; *Haworth et al., 2007*), we found that *Shh* mRNA expression in the mandibular arch epithelium is restricted to the oropharyngeal side at E10.5 (*Figure 3E and E'*). Corresponding to this restricted *Shh* expression pattern, expression of *Ptch1, Foxf1,* and *Foxf2,* known transcriptional target genes of hedgehog signaling (*Hoffmann et al., 2014*; *Jeong et al., 2004*), is restricted to the oropharyngeal side of the mandibular arch cells underneath the *Shh*-expressing epithelium (*Figure 3F–H* and *Figure 3F'–H'*). Whereas *Bmp4* and *Fgf8* mRNAs are expressed in the distal and proximal regions of the oral ectoderm (*Figure 3I,I', M,M', N,N'*), respectively, the down-stream target genes of BMP

signaling, including *Msx1*, *Msx2* and *Alx4,* are expressed in an aboral-to-oropharyngeal gradient pattern complementary to that of Foxf1 and Foxf2 in the distal mandibular arch mesenchyme (*Figure 3J–L*, and *Figure 3J'–L'*, compare with *Figure 3G'–H'*). In contrast, the *Lhx8* and *Lhx6* mRNAs are expressed in a limited region of the rostral mandibular mesenchyme underlying the *Fgf8* expressing epithelium (*Figure 3O–P* and *Figure 3O'–P'*). These results demonstrate clearly that the rostral-caudal axis of the mandibular arch mesenchyme along which the *Fgf8-Lhx6/8* pathway is activated is distinct from the oral-aboral axis along which the Hedgehog and BMP signaling pathways exhibit complementary activity in the mandibular arch mesenchyme. These results also highlight the clarity of visualizing differential gene expression patterns along the oral-aboral axis using sagittal sections and the limitation of frontal views (compare *Figure 3B,C* with *Figure 3G,K*) or frontal sections for analysis of gene expression patterns in the developing facial processes.

## $Smo^{c/c}$;*Wnt1-Cre* mouse embryos exhibit defects in oral-aboral patterning of the mandible

To investigate whether hedgehog signaling plays a direct role in patterning the mandibular arch mesenchyme along the oral-aboral axis, we first examined $Smo^{c/c}$;*Wnt1-Cre* mouse embryos in which Smo, the obligatory and cell-autonomous transducer of hedgehog signaling (*Briscoe and Vincent, 2013*; *Jeong et al., 2004*; *Zhang et al., 2001*), is inactivated in all cranial neural crest cells (*Danielian et al., 1998*; *Jeong et al., 2004*). Similar to the phenotype of $Smo^{c/n}$;*Wnt1-Cre* mice ($Smo^n$ is a *Smo*-null allele) as previously reported (*Jeong et al., 2004*), the $Smo^{c/c}$;*Wnt1-Cre* mice die perinatally and exhibit severe truncation of facial structures, including the maxilla and mandible (*Figure 4A–B*), and agenesis of the oral tongue tissues (*Figure 4—figure supplement 1*). Examination of the mandibular skeleton in E18.5 mouse embryos showed that the $Smo^{c/c}$;*Wnt1-Cre* mutants had mirror-image duplication of the dentary bone, including duplication of the condylar process, although the mandibular skeleton is distally truncated (*Figure 4C–D*). A similar partial duplication of the dentary was reported in the $Smo^{c/n}$;*Wnt1-Cre* mutants (*Jeong et al., 2004*) but the tissue source of the duplicated dentary bone was not investigated. We analyzed coronal sections of E13.5 $Smo^{c/c}$; *Wnt1-Cre* embryos and their littermates and found that the $Smo^{c/c}$;*Wnt1-Cre* embryos had ectopic ossification in the oral side of the mandibular mesenchyme where the tongue failed to form, as indicated by ectopic expression of Runx2 and alkaline phosphatase in the oral side of the mandibular mesenchyme (*Figure 4E–H*). These results suggest that hedgehog signaling in the neural crest cells is required for proper specification of the oral side of the mandibular mesenchyme.

We compared the patterns of expression of the BMP target genes *Msx1*, *Msx2*, and *Alx4* in the $Smo^{c/c}$;*Wnt1-Cre* mutant embryos and their control littermates. We found that expression of *Msx1*, *Msx2*, and *Alx4* mRNAs were all expanded to the oropharyngeal side of the distal mandibular arch mesenchyme in the $Smo^{c/c}$;*Wnt1-Cre* mutant embryos by E10.5 (*Figure 4—figure supplement 2*), suggesting that hedgehog signaling might specify oral fate of the mandibular mesenchyme through antagonizing BMP signaling.

## Hedgehog signaling in the postmigratory neural crest-derived mandibular mesenchyme is required for oral-aboral patterning of the mandible

It has been reported that the $Smo^{c/n}$;*Wnt1-Cre* mouse embryos had significantly increased apoptosis of neural-crest derived facial mesenchyme, including mandibular arch mesenchyme at E9.5 to E10.5 (*Jeong et al., 2004*). One possible explanation of the dentary bone duplication and altered patterns of gene expression we have observed in the $Smo^{c/c}$;*Wnt1-Cre* mutant embryos is that those defects might be secondary to altered neural crest migration and/or cell death caused by lack of hedgehog signaling in the migrating neural crest cells. To investigate further whether hedgehog signaling plays a direct role in oral-aboral patterning of the mandibular mesenchyme, we generated and analyzed mandibular development in $Smo^{c/c}$;*Hand2-Cre* mice, in which Cre expressed from the *Hand2-Cre* transgene is activated in post-migratory neural crest cells in the distal halves of the mandibular and other pharyngeal arches (*Ruest et al., 2003*). By crossing the *Hand2-Cre* transgenic mice with the *R26R-lacZ* Cre reporter mouse line (*Soriano, 1999*) and analyzing LacZ activity in the *R26R-lacZ*; *Hand2-Cre* embryos, we confirmed that Cre expressed from the *Hand2-Cre* transgene starts to be active after E9.0 and is restricted in the mandibular and other pharyngeal arches (*Figure 5—figure*

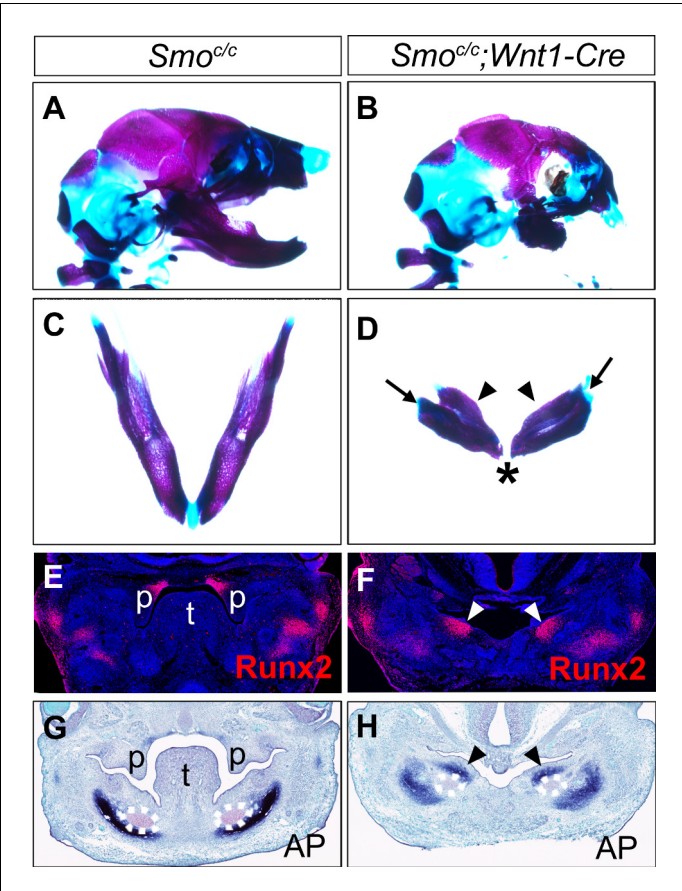

**Figure 4.** $Smo^{c/c}$;Wnt1-Cre embryos display tongue agenesis and ectopic ossification. (**A, B**) Lateral view of skeletal preparations of heads of E18.5 $Smo^{c/c}$ (**A**) and $Smo^{c/c}$;Wnt1-Cre (**B**) embryos (n = 3 for each genotype). (**C, D**) Comparison of the mandibular skeletons of E18.5 $Smo^{c/c}$ (**C**) and $Smo^{c/c}$;Wnt1-Cre (**D**) embryos (n = 3 for each genotype). Arrows in D point to condylar cartilage and the arrowheads point to the duplicated dentary bone. Asterisk in D indicates truncation of the distal mandible in $Smo^{c/c}$;Wnt1-Cre embryos. (**E, F**) Immunofluorescent detection of Runx2 protein (red) on frontal sections of E12.5 $Smo^{c/c}$ (**E**), and $Smo^{c/c}$;Wnt1-Cre (**F**) embryos (n = 3 for each genotype). Sections were counterstained with DAPI. (**G, H**) Detection of alkaline phosphatase activity (blue) on frontal sections of E13.5 $Smo^{c/c}$ (**G**) and $Smo^{c/c}$;Wnt1-Cre (**H**) embryos (n = 3 for each genotype). Arrowheads in F and H point to the ectopic ossification at the oral side of mandibular mesenchyme. The condensed Meckel's cartilage primordia are outlined using dashed lines in G and H. p, palate; t, tongue.
DOI: https://doi.org/10.7554/eLife.40315.014

The following figure supplements are available for figure 4:

**Figure supplement 1.** $Smo^{c/c}$;Wnt1-Cre display agenesis of the oral tongue structure.
DOI: https://doi.org/10.7554/eLife.40315.015

**Figure supplement 2.** Mouse embryos with tissue-specific inactivation of Smo signaling in the neural crest lineage exhibit altered pattern of expression of BMP target genes in the mandibular arch.
DOI: https://doi.org/10.7554/eLife.40315.016

supplement 1). From E10.5 to E12.5, the *Hand2-Cre* lineage cells contribute to about the distal two thirds of the mandibular mesenchyme, including neural crest-derived mesenchyme in the developing tongue (*Figure 5—figure supplement 1B–D,F–H*), similar to previous findings (*Ruest et al., 2003*; *Ruest et al., 2004*). We then analyzed the effect of tissue-specific inactivation of *Smo* using the Hand2-Cre driver on the gene expression patterns in the distal mandibular arch mesenchyme. As shown in *Figure 5*, whereas expression of *Bmp4* mRNAs was similarly restricted to the distal mandibular ectoderm in the $Smo^{c/c}$ control and $Smo^{c/c}$;Hand2-Cre mutant embryos at E10.5 (*Figure 5A,B*), the pattern of expression of *Msx1*, *Msx2*, and *Alx4*, is each expanded to the oropharyngeal side of the distal mandibular arch mesenchyme in the $Smo^{c/c}$;Hand2-Cre mutant embryos compared with

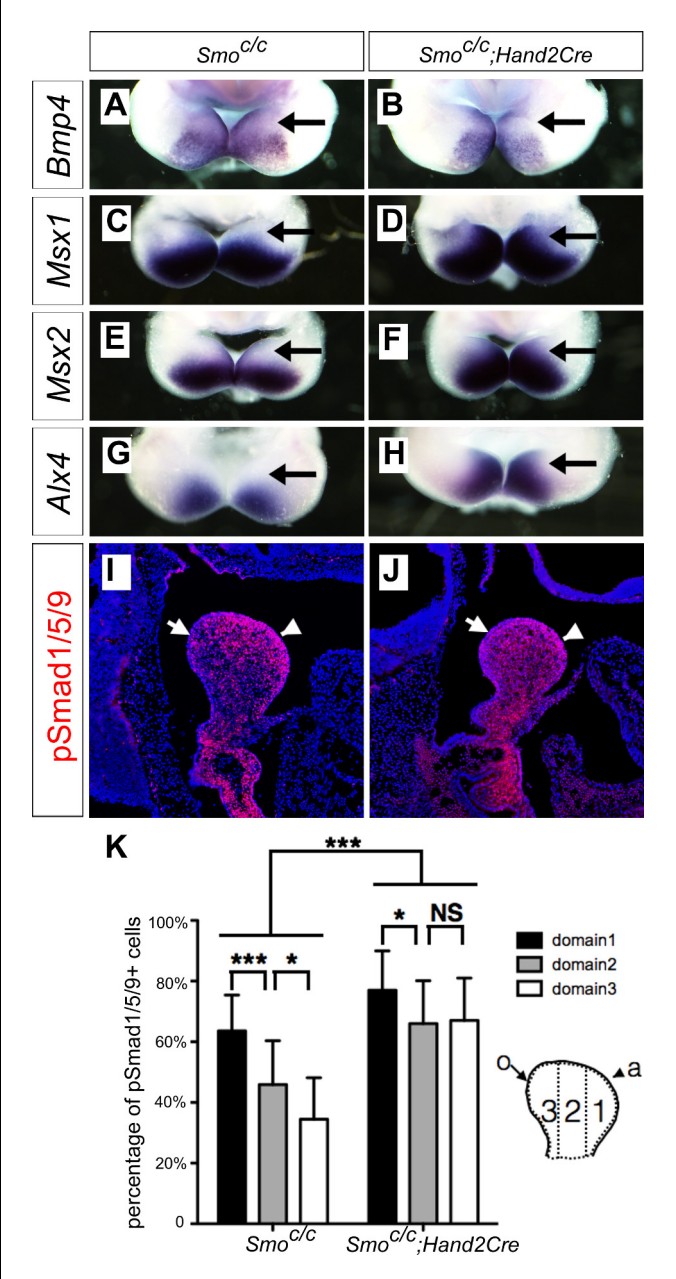

**Figure 5.** *Smo^{c/c};Hand2-Cre* embryos exhibit oropharyngeally expanded BMP signaling activity in the developing mandibular arch. (A–H) Whole mount in situ hybridization detection of *Bmp4* (A, B), *Msx1* (C, D), *Msx2* (E, F), and *Alx4* (G, H) mRNAs in the mandibular arches in E10.5 *Smo^{c/c}* (A, C, E, G), and *Smo^{c/c};Hand2-Cre* (B, D, F, H) embryos (n = 3 for each genotype). (I, J) Immunofluorescent detection of phospho-Smad1/5/9 (pSmad1/5/9, red) on sagittal sections through the distal region of the mandibular arches in E10.5 *Smo^{c/c}* (I) and *Smo^{c/c};Hand2-Cre* (J) embryos . Arrow points to oropharyngeal side and arrowhead points to aboral side of the mandibular arch. (K) Quantification of percentage of pSmad1/5/9 positive nuclei in the three domains along the aboral-oral axis of mandibular mesenchyme (domain1, domain2, domain3). Statistical analysis was performed on data from 22 sections of 4 embryos of each genotype by using two-way ANOVA. a, aboral, o, oral. *$p<0.05$, ***$p<0.001$.
DOI: https://doi.org/10.7554/eLife.40315.017

The following figure supplement is available for figure 5:

**Figure supplement 1.** Tissue specificity of Cre-mediated activation of lacZ expression in the developing craniofacial complex of *Hand2-Cre;R26R-LacZ* embryos from E9.5 to E12.5.
DOI: https://doi.org/10.7554/eLife.40315.018

the *Smo^{c/c}* control embryos (*Figure 5C–H*). Furthermore, we found by immunofluorescent staining that phosphorylated Smad1/5/9 (pSmad1/5/9) positive cells exhibit a gradient distribution, with the highest level in the mandibular mesenchyme immediately underlying the *Bmp4*-expressing mandibular ectoderm and the lowest level at the pharyngeal side of the mandibular arch mesenchyme, in E10.5 control embryos, but the E10.5 *Smo^{c/c};Hand2-Cre* mutant embryos exhibited high levels of pSmad1/5/9 throughout the oral-aboral axis of the distal mandibular arch mesenchyme (*Figure 5I–K*).

We further analyzed the phenotypic effects in the *Smo^{c/c};Hand2-Cre* mutant embryos. Skeletal preparations of E18.5 embryos showed that the *Smo^{c/c};Hand2-Cre* embryos had a shortened mandible but their skull and upper jaw were mostly unaffected, in comparison with the control littermates (*Figure 6A,B*). Histological analysis showed that the *Smo^{c/c};Hand2-Cre* embryos displayed severe defect in tongue formation although they had normal development of the nasal and upper jaw structures, including normal development of the secondary palate (*Figure 6C,D*). Similar to that reported in *Smo^{c/n};Wnt1-Cre* embryos (*Jeong et al., 2004*), *Myf5*-expressing tongue muscle precursor cells arriving at the mandibular arch in the E10.5 *Smo^{c/c};Hand2-Cre* embryos were significantly reduced compared with the control littermates (*Figure 6—figure supplement 1A,B*). At E15.5, the *Smo^{c/c}; Hand2-Cre* mutant embryos displayed a cleft and rudimentary tongue in the pharyngeal region and absence of oral part of the tongue (*Figure 6—figure supplement 1C–H*). Analysis of expression of Runx2 and alkaline phosphatase on coronal sections showed that the *Smo^{c/c};Hand2-Cre* mutant embryos exhibited ectopic ossification in the oral side of the mandibular mesenchyme as early as E12.5 (*Figure 6E–H*). These results indicate that inactivation of *Smo* in the neural crest cells either prior to their migration into the mandibular arch, as in *Smo^{c/c};Wnt1-Cre* embryos, or after their arrival in the mandibular arch, as in *Smo^{c/c};Hand2-Cre* embryos, led to similar disruption of mandibular structures, including disruption of tongue formation and ectopic bone formation from the oral side of the mandibular arch mesenchyme. Together, these results indicate that hedgehog signaling in the neural crest-derived mandibular mesenchyme is crucial for inducing tongue formation and preventing osteogenic differentiation in the oral side of the developing mandible.

## Ectopic activation of hedgehog signaling in the mandibular arch mesenchyme inhibits BMP signaling pathway activation

The oropharyngeally expanded activation of pSmad1/5/9 and expression of the BMP targeted genes in the mandibular arch mesenchyme in *Smo^{c/c};Hand2-Cre* mutant embryos suggest that hedgehog signaling patterns the oral-aboral axis of the mandibular mesenchyme through antagonizing BMP signaling. To test this hypothesis further, we crossed *Hand2-Cre* transgenic mice with the *R26SmoM2* mice, which carry a Cre-activatable transgene encoding a dominant active form of Smo (*Jeong et al., 2004*), and analyzed the effects of ectopic activation of Smo-mediated hedgehog signaling in the mandibular mesenchyme in *R26SmoM2;Hand2-Cre* mouse embryos. As shown in *Figure 7*, expression of *Foxf1*, a known direct transcriptional target gene downstream of hedgehog signaling (*Hoffmann et al., 2014*), is activated throughout the oral-aboral axis of the distal mandibular arch mesenchyme in E10.5 *R26SmoM2;Hand2-Cre* embryos, in comparison with the restricted expression of *Foxf1* at the oropharyngeal side of the mandibular arch mesenchyme in control littermates (*Figure 7A,B*). Compared with the control littermates, expression of *Msx2* and *Alx4* in the distal mandibular arch mesenchyme is significantly reduced in the E10.5 *R26SmoM2;Hand2-Cre* mouse embryos (*Figure 7C–F*). Furthermore, immunofluorescent staining of sagittal sections of E10.5 embryos showed a significant decrease in the number of pSmad1/5/9 positive cells in the distal mandibular arch mesenchyme of E10.5 *R26SmoM2;Hand2-Cre* embryos compared with control littermates (*Figure 7G–I*). These results confirm that hedgehog-Smo signaling regulates BMP signaling pathway activation along the oral-aboral axis in the mandibular arch mesenchyme.

## Role of BMP signaling in hedgehog-mediated regulation of neural crest survival in the mandibular arch

Previous studies showed that blocking Smo-mediated hedgehog signaling causes increased apoptosis of the neural crest cells in the developing facial primordia in both chick and mouse embryos (*Ahlgren and Bronner-Fraser, 1999*; *Hu and Helms, 1999*; *Jeong et al., 2004*). The truncation of the maxillary and mandibular structures in *Smo^{c/n};Wnt1-Cre* mutant mice was partly attributed to

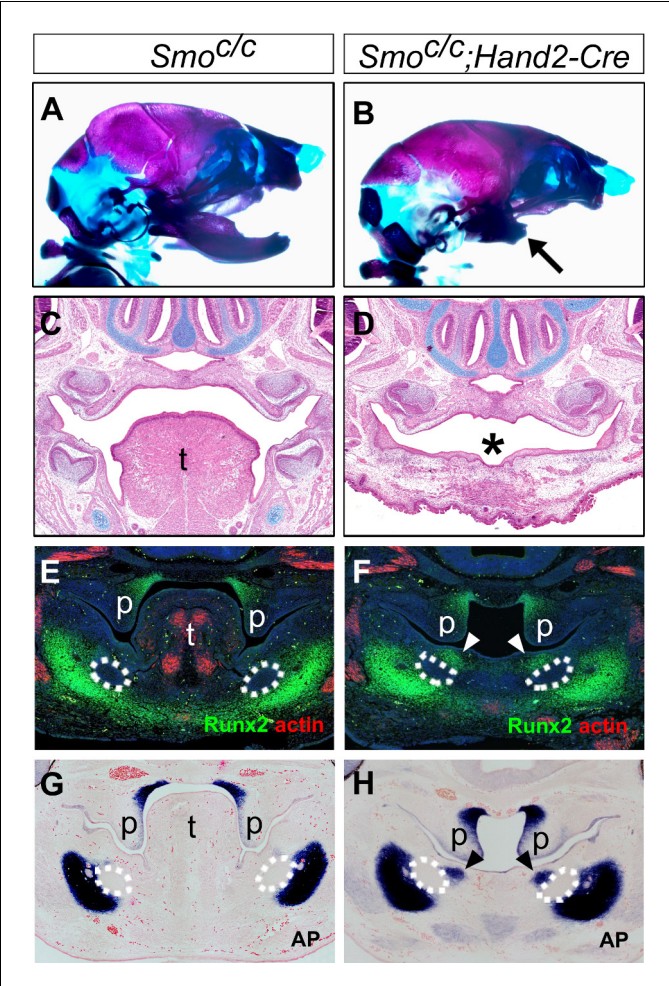

**Figure 6.** $Smo^{c/c};Hand2-Cre$ embryos display tongue agenesis and ectopic ossification. (**A, B**) Skeletal preparations of heads of E18.5 $Smo^{c/c}$ (**A**) and $Smo^{c/c};Hand2-Cre$ (**B**) embryos (n = 5 for each genotype). Arrow in B indicates the shortened mandible in the $Smo^{c/c};Hand2-Cre$ embryo. (**C, D**) HE staining of frontal sections of E16.5 $Smo^{c/c}$ (**C**) and $Smo^{c/c};Hand2-Cre$ (**D**) embryos (n = 3 for each genotype). Asterisk in D marks absence of tongue in the $Smo^{c/c};Hand2-Cre$ embryo. (**E, F**) Immunofluorescent detection of Runx2 (green) and muscle alpha-actin (red) on frontal sections of E12.5 $Smo^{c/c}$ (**E**) and $Smo^{c/c};Hand2-Cre$ (**F**) embryos (n = 3 for each genotype). Sections were counterstained with DAPI. (**G, H**) Detection of alkaline phosphatase activity (blue) on frontal sections of E12.5 $Smo^{c/c}$ (**G**) and $Smo^{c/c};Hand2-Cre$ (**H**) embryos (n = 3 for each genotype). Sections were counterstained with eosin. Arrowheads in F and H indicate the ectopic ossification at the oral side of the developing mandible. The condensed Meckel's cartilage primordia are outlined using dashed lines in E-H. p, palate; t, tongue.
DOI: https://doi.org/10.7554/eLife.40315.019

The following figure supplement is available for figure 6:

**Figure supplement 1.** $Smo^{c/c};Hand2-Cre$ display agenesis of the oral tongue structure.
DOI: https://doi.org/10.7554/eLife.40315.020

increased apoptosis of neural crest cells in the mutant facial primordia at E9.5 to E10.5 (*Jeong et al., 2004*). We found that $Smo^{c/c};Hand2-Cre$ mutant pups also exhibit significantly shortened mandible compared with control littermates (*Figure 6A,B*). We therefore analyzed cell apoptosis in $Smo^{c/c};Hand2-Cre$ mutant embryos and their littermates using immunofluorescent detection of active Capase-3. Whereas no increase in apoptosis was detectable in E9.5 $Smo^{c/c};Hand2-Cre$ embryos, we detected a group of cells with strong Caspase-3 activity specifically in the distal mandibular arch mesenchyme in E10.5 $Smo^{c/c};Hand2-Cre$ embryos (*Figure 8A,B*). The increased apoptosis in the distal mandibular mesenchyme correlated with the distal truncation of both the dentary

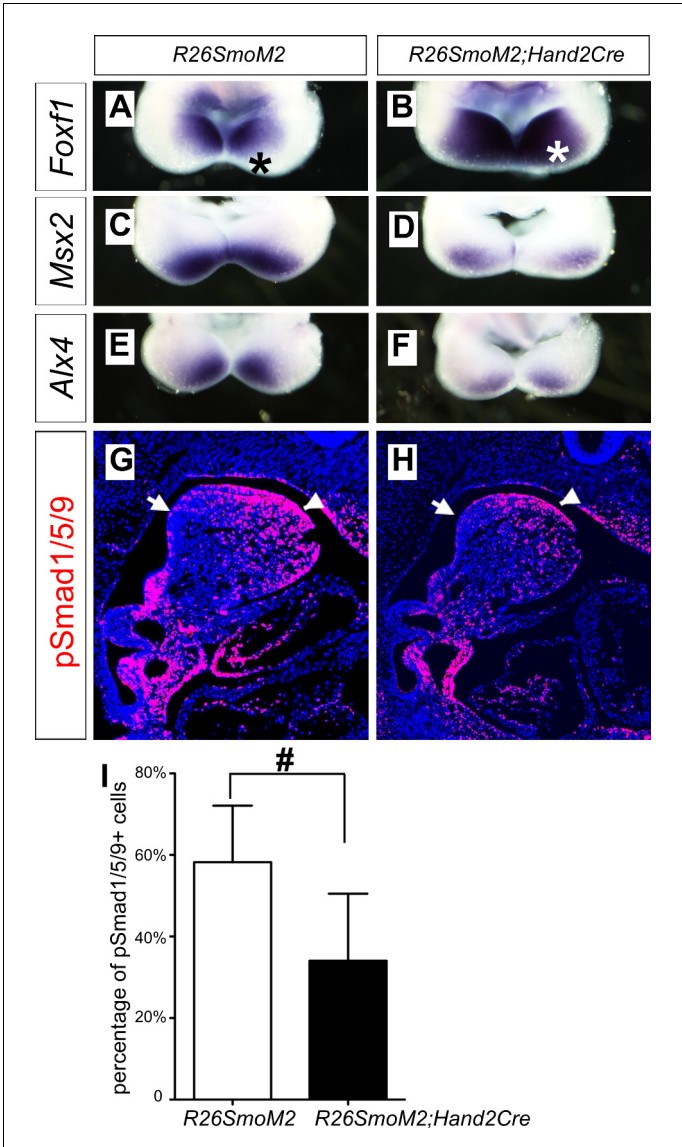

**Figure 7.** Ectopic activation of Smo signaling inhibits BMP signaling activity in the mandibular arch mesenchyme. (A–F) Rostral views of the mandibular arches following whole mount in situ hybridization detection of *Foxf1* (A, B), *Msx2* (C, D), and *Alx4* (E, F) mRNAs in the E10.5 *R26SmoM2* (A, C, E), and *R26SmoM2;Hand2-Cre* (B, D, F) embryos (n = 3 for each genotype). Asterisks in A and B mark the aboral side of the mandibular arch. (G, H) Immunofluorescent detection of phospho-Smad1/5/9 (pSmad1/5/9, red) on sagittal sections through the distal regions of E10.5 *R26SmoM2* (G) and *R26SmoM2;Hand2-Cre* (H) embryos (n = 3 for each genotype). Arrow points to oropharyngeal endoderm, and arrowhead points to mandibular ectoderm. (I) Quantification of the percentage of pSmad1/5/9 positive nuclei in the mandibular arch mesenchyme. Statistical analysis was performed on data from 18 sections of three embryos for each genotype by using Student's t-test. #, p<0.001.
DOI: https://doi.org/10.7554/eLife.40315.021

bone on the aboral side and the ectopic bone on the oral side of the mutant mandible (*Figure 8E, F*). Since we detected expanded activation of pSmad1/5/9 and Bmp4 target gene expression in the mandibular arch mesenchyme in the E10.5 $Smo^{c/c}$;Hand2-Cre mutant embryos and since increased BMP signaling has been correlated with increased neural crest apoptosis in mouse embryos lacking both Noggin and Chordin, two endogenous antagonists of BMPs (*Anderson et al., 2006*; *Stottmann et al., 2001*), we investigated whether reducing the *Bmp4* gene dosage could rescue mandibular morphogenesis in the $Smo^{c/c}$;Hand2-Cre embryos. We found that the amount of active

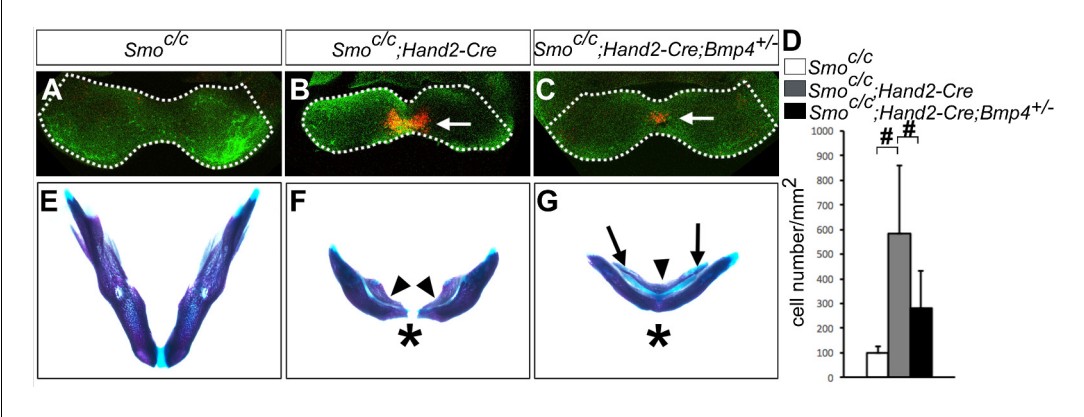

**Figure 8.** *Smo^c/c^;Hand2-Cre* embryos showed Bmp4-dependent increase in apoptosis of distal mandibular mesenchyme. (**A–C**) Immunofluorescent detection of active-Caspase3 (red) in E10.5 *Smo^c/c^* (**A**), *Smo^c/c^;Hand2-Cre* (**B**), and *Smo^c/c^;Hand2-Cre;Bmp4^+/-^* (**C**) embryos. Mandibles are outlined with white dash lines. (**D**) Quantification of active-Caspase3 positive cell density in the *Smo^c/c^*, *Smo^c/c^;Hand2-Cre* and *Smo^c/c^;Hand2-Cre;Bmp4^+/-^* samples. The results were presented as mean ± SD. *Smo^c/c^* (98.96 ± 28.21 cells/mm²); *Smo^c/c^;Hand2-Cre* (584.12 ± 275.20 cells/mm²); *Smo^c/c^;Hand2-Cre;Bmp4^+/-^* (280.45 ± 153.23 cells/mm²). Analysis was performed with one-way ANOVA followed by Tukey post hoc test to compare all pairs. #, p<0.001. (**E–G**) Skeletal preparations showing mandibles of E18.5 *Smo^c/c^* (**E**), *Smo^c/c^;Hand2-Cre* (**F**), and *Smo^c/c^;Hand2-Cre;Bmp4^+/-^* (**G**) embryos (n = 5 for each genotype). Arrowheads in F and G point to duplicated dentary bone. Asterisks in F and G mark differences in the morphology of the distal mandibular structure in *Smo^c/c^;Hand2-Cre* (**F**) and *Smo^c/c^;Hand2-Cre;Bmp4^+/-^* (**G**) embryos. Arrows in G point to the Meckel's cartilage.

DOI: https://doi.org/10.7554/eLife.40315.022

caspase-3 positive cells was significantly reduced in the mandibular mesenchyme of E10.5 *Smo^c/c^;Hand2-Cre;Bmp4^+/-^* embryos in comparison with *Smo^c/c^;Hand2-Cre* littermates (*Figure 8B–D*). By E18.5, 4 out of 5 *Smo^c/c^;Hand2-Cre;Bmp4^+/-^* mutants showed partial rescue of distal mandible and formation of the mandibular symphysis, in contrast to the lack of distal mandibular skeletal structures in the *Smo^c/c^;Hand2Cre* mutants (*Figure 8F,G*). However, the *Smo^c/c^;Hand2-Cre;Bmp4^+/-^* pups still showed tongue agenesis and duplication of the dentary bone on the oral side (*Figure 8G*). These results indicate that hedgehog signaling regulates mandibular arch mesenchyme survival and oral-aboral patterning through antagonizing BMP signaling.

## Foxf1 and Foxf2 mediate hedgehog signaling function in patterning the oral-aboral axis of the mandible

It has been shown previously that hedgehog signaling regulates expression of several members of the *Fox* gene family, including *Foxf1* and *Foxf2*, in the developing mouse facial primordia (*Jeong et al., 2004*). We found that expression of *Foxf1* and *Foxf2*, as well as *Ptch1* mRNAs, was dramatically reduced in the distal mandibular arch mesenchyme in E10.5 *Smo^c/c^;Hand2-Cre* embryos compared with their control littermates (*Figure 9A–F*). We recently showed that neural crest-specific inactivation of either *Foxf1* or *Foxf2* caused cleft palate but did not significantly affect mandible and tongue development (*Xu et al., 2016b*). We next tested the possibility that Foxf1 and Foxf2 act redundantly to mediate hedgehog signaling function in mandibular development by generating and analyzing *Foxf1^c/c^;Foxf2^c/c^;Wnt1-Cre* mouse embryos. Remarkably, we found that the *Foxf1^c/c^;Foxf2^c/c^;Wnt1-Cre* mouse embryos exhibited absence of the oral tongue and ectopic ossification at the oral side of the mandibular mesenchyme (*Figure 9G–L*), similar to *Smo^c/c^;Wnt1-Cre* (*Figure 4E–H*) and *Smo^c/c^;Hand2-Cre* (*Figure 6E–H*) embryos. In contrast to the severe truncation of distal mandibular skeleton in the *Smo^c/c^;Wnt1-Cre* (*Figure 4B,D*) and *Smo^c/c^;Hand2-Cre* (*Figure 6B*, *Figure 8F*) mutant pups, however, the E18.5 *Foxf1^c/c^;Foxf2^c/c^;Wnt1-Cre* mutant pups showed mandibular bone with nearly normal symphysis at the distal end and with a thin layer of ectopic bone formed at the oral side (*Figure 9K,L*). In addition, migration of the *Myf5*-expressing tongue muscle precursor cells to the E10.5 mandibular arch occurred normally in the *Foxf1^c/c^;Foxf2^c/c^;Wnt1-Cre* mutant embryos, similar to control littermates (*Figure 9—figure supplement 1A,B*) and unlike the *Smo^c/c^;Hand2-Cre* embryos (*Figure 6—figure supplement 1B*). Careful examination of muscular structures by immunofluorescent detection using serial sections of the E16.5 embryos showed that

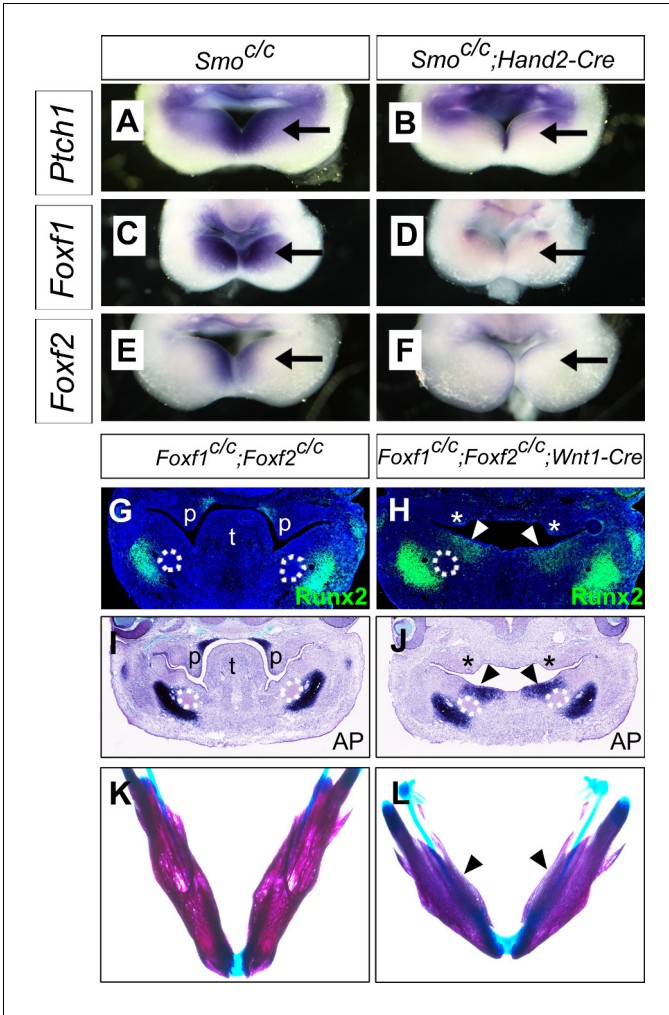

**Figure 9.** Foxf1 and Fox2 mediate Shh-Smo signaling function in mandible and tongue development. (A–F) Rostral views of the mandibular arches of E10.5 $Smo^{c/c}$ (A, C, E) and $Smo^{c/c};Hand2-Cre$ (B, D, F) embryos following whole mount in situ hybridization detection of $Ptch1$ (A, B), $Foxf1$ (C, D), and $Foxf2$ (E, F) mRNAs (n = 3 for each genotype). Arrow points to the mRNA signals in the oral side of the mandibular arch. (G, H) Immunofluorescent detection of Runx2 (green) on frontal sections of E12.5 $Foxf1^{c/c};Foxf2^{c/c}$ (G) and $Foxf1^{c/c};Foxf2^{c/c};Wnt1-Cre$ (H) embryos (n = 3 for each genotype). Sections were counterstained with DAPI. (I, J) Detection of alkaline phosphatase activity (blue) on frontal sections of E12.5 $Foxf1^{c/c};Foxf2^{c/c}$ (I) and $Foxf1^{c/c};Foxf2^{c/c};Wnt1-Cre$ (J) embryos (n = 3 for each genotype). (K, L) Skeletal preparations showing mandibles of E18.5 $Foxf1^{c/c};Foxf2^{c/c}$ (K) and $Foxf1^{c/c};Foxf2^{c/c};Wnt1-Cre$ (L) embryos. Arrowheads in H, J, and L point to ectopic ossification at the oral side of the mandible (n = 3 for each genotype). Asterisks in H and J mark the defective palatal shelves in $Foxf1^{c/c};Foxf2^{c/c};Wnt1-Cre$ embryos. The Meckel's cartilage primordia are indicated by white dashed circles in G-J. p, palate; t, tongue.

DOI: https://doi.org/10.7554/eLife.40315.023

The following figure supplement is available for figure 9:

**Figure supplement 1.** $Foxf1^{c/c}; Foxf2^{c/c};Wnt1-Cre$ display agenesis of the oral tongue structure.

DOI: https://doi.org/10.7554/eLife.40315.024

the $Foxf1^{c/c};Foxf2^{c/c};Wnt1-Cre$ mutant embryos formed extensive extrinsic tongue muscles but only rudimentary intrinsic tongue muscles in the pharyngeal region (*Figure 9—figure supplement 1C–H*). We further examined pSmad1/5/9 activation in the mandibular arches in E10.5 $Foxf1^{c/c};Foxf2^{c/c};$ $Wnt1-Cre$ embryos and their littermates and found that the $Foxf1^{c/c};Foxf2^{c/c};Wnt1-Cre$ embryos displayed expansion of pSmad1/5/9 activation to the oropharygeal side of the distal mandibular arch mesenchyme (*Figure 10*), similar to E10.5 $Smo^{c/c};Hand2-Cre$ mouse embryos (*Figure 5I–K*). These

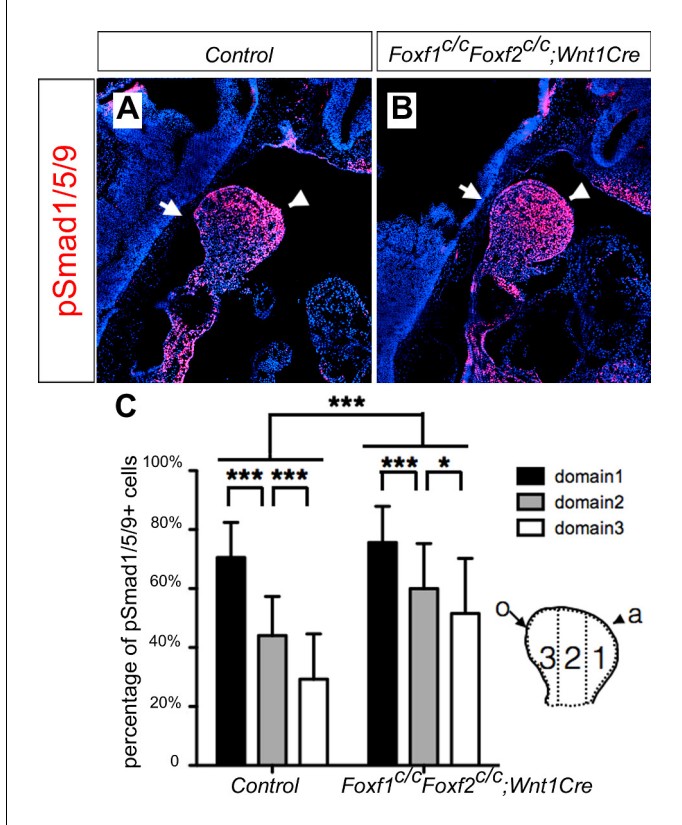

**Figure 10.** *Foxf1^{c/c};Foxf2^{c/c};Wnt1-Cre* embryos exhibit oropharyngeally expanded BMP signaling activity in the developing mandibular arch. (**A, B**) Immunofluorescent detection of phospho-Smad1/5/9 (pSmad1/5/9, red) on sagittal sections through the distal region of the mandibular arches in E10.5 *control* (littermates that are Cre negative) (**A**) and *Foxf1^{c/c};Foxf2^{c/c};Wnt1-Cre* (**B**) embryos. Arrow points to oropharyngeal side and arrowhead points to aboral side of the mandibular arch. (**C**) Quantification of the percentage of pSmad1/5/9 positive nuclei in the three domains along aboral-oral axis of mandibular arch mesenchyme (domain1, domain2, domain3). Statistical analysis was performed on data from 28 sections of 5 control embryos and 33 sections of 5 *Foxf1^{c/c};* *Foxf2^{c/c};Wnt1-Cre* embryos by using two-way ANOVA. a, aboral, o, oral. ***$p<0.001$.
DOI: https://doi.org/10.7554/eLife.40315.025

results indicate that Foxf1 and Foxf2 are important mediators of hedgehog signaling function in the mandibular arch mesenchyme for tongue formation and for preventing ossification of the oral side mandibular arch mesenchyme.

## Discussion

Prior to this report, only Fgf8 signaling has been directly implicated in patterning the oral-aboral axis in vertebrates (*Cobourne and Sharpe, 2003*; *Grigoriou et al., 1998*; *Tucker et al., 1999*). *Grigoriou et al. (1998)* first showed by using frontal sections of E10.5 mouse embryos that expression of *Lhx6* and *Lhx8* (named *Lhx7* in the original publication) mRNAs in the developing facial primordia was asymmetrically localized to caudal side of the maxillary processes and rostral side of the mandibular processes and that Fgf8, but not Bmp4 or Shh, was able to induce *Lhx6/8* mRNA expression in E10.5 mouse embryonic mandibular mesenchyme explants in culture (*Grigoriou et al., 1998*). *Tucker et al., 1999* showed that Fgf8 could not only induce *Lhx6/8* expression but also repress expression of *Goosecoid* (*Gsc*), a homeobox gene normally expressed in the caudal side of the mandibular arch mesenchyme in E10.5 mouse embryos, in embryonic mouse mandibular explants (*Tucker et al., 1999*). Further studies showed that Fgf8-mediated repression of *Gsc* expression in the mandibular mesenchyme explants depended on induction of *Lhx6/8*, suggesting that Fgf8, normally expressed in the rostroproximal mandibular arch epithelium, patterns the rostral-caudal axis of

the mandibular mesenchyme by inducing expression of *Lhx6/8* in the rostral mandibular mesenchyme and secondarily restricting *Gsc* expression to the caudal mandibular mesenchyme (*Tucker et al., 1999*). Whereas these reports and subsequent review articles have equated the rostral-caudal axis of the mandibular arch as the same as the oral-aboral axis (*Chai and Maxson, 2006*; *Cobourne and Sharpe, 2003*; *Grigoriou et al., 1998*; *Parada and Chai, 2015*; *Tucker et al., 1999*), both our scRNA-seq data and our in situ hybridization data comparing the expression patterns of *Fgf8*, *Lhx6*, and *Lhx8*, with the expression patterns of the hedgehog pathway genes *Shh*, *Ptc1*, *Foxf1*, *Foxf2*, and the BMP pathway genes *Bmp4*, *Msx1*, *Msx2*, in the developing mouse mandibular arch (*Figure 3*) clearly demonstrate that the hedgehog and BMP pathway genes are expressed in a complementary pattern along the oral-aboral axis of the mandibular arch that is distinct from the rostral-caudal axis along which the *Fgf8-Lhx6/8* gene pathway is activated. Furthermore, our data showing that conditional inactivation of *Smo* in either the early neural crest lineage or specifically in the post-migratory neural crest-derived distal mandibular arch mesenchyme led to agenesis of the oral tongue structure and duplication of the dentary bone. Although partial duplication of the dentary was reported previously in the $Smo^{c/n}$;*Wnt1-Cre* mutants, most of the neural crest derived craniofacial bones were missing or severely hypoplastic in the $Smo^{c/n}$;*Wnt1-Cre* mutants (*Jeong et al., 2004*) and the tissue source of the partially duplicated dentary was not investigated. We show that the duplicated dentary bone in the $Smo^{c/c}$;*Wnt1-Cre* and $Smo^{c/c}$;*Hand2-Cre* mutants form from ectopic ossification in the oral side of the mandibular arch mesenchyme and the ectopic dentary bone formation was preceded by expansion of BMP signaling activation into the oral region of the mandibular mesenchyme, demonstrating clearly for the first time that hedgehog signaling in the mandibular arch mesenchyme is required for patterning the oral-aboral axis of the mandible.

Hedgehog signaling, particularly Shh signaling, has been shown to regulate a number of tissue patterning events during vertebrate development, including anteroposterior patterning of the limb, dorsoventral patterning of the neural tube, and dorsoventral patterning of somites (*Briscoe, 2009*; *Ericson et al., 1997*; *Hirsinger et al., 1997*; *Ingham and McMahon, 2001*; *McMahon et al., 1998*; *Riddle et al., 1993*; *Tickle and Towers, 2017*; *Watanabe et al., 1998*). In several of these developmental processes, Shh signaling regulates tissue patterning through antagonizing BMP signaling. During limb development, Shh signaling controls expression of *Gremlin1*, which encodes a BMP antagonist that is crucial for maintaining the apical ectodermal ridge and the Shh-Fgf4/8 positive feedback loop for limb outgrowth (*Khokha et al., 2003*; *Zúñiga et al., 1999*). In the developing neural tube, cells adopt distinct identities along the dorsoventral axis in response to Shh from the notochord and floor plate as well as to BMPs produced in the overlying ectoderm and roof plate (*Ericson et al., 1997*; *Lee and Jessell, 1999*). Whereas Shh signaling is essential for inducing ventral cell fates in the developing neural tube, studies of *Noggin* mutant mice showed that Noggin-mediated antagonism of BMP signaling is required for Shh-mediated ventralization of the spinal cord (*McMahon et al., 1998*). Similarly, during somite development, Noggin-mediated antagonism of BMP signaling is required for Shh-mediated induction and dorsoventral patterning of the sclerotomal cells (*Hirsinger et al., 1997*; *McMahon et al., 1998*). In this study, we found that expression of several downstream target genes of the hedgehog and BMP pathways are expressed in complementary patterns along the oral-aboral axis of the mandibular arch mesenchyme, which correlated with the complementary patterns of expression of *Shh* and *Bmp4* in the distal mandibular arch epithelium. We found that expression of the BMP target genes, including *Msx1*, *Msx2*, and *Alx4*, expanded to the oropharyngeal side of the mandibular arch mesenchyme in the $Smo^{c/c}$;*Wnt1-Cre* and $Smo^{c/c}$; *Hand2-Cre* embryos. Furthermore, we found that pSmad1/5/9 was activated throughout the oral-aboral axis of the distal mandibular arch mesenchyme in $Smo^{c/c}$;*Hand2-Cre* embryos, indicating that hedgehog signaling antagonizes BMP signaling pathway activation in the oropharyngeal side of the mandibular arch mesenchyme. We found that conditional deletion of *Foxf1* and *Foxf2*, two of several Fox family genes with Smo-dependent expression in the mandibular arch mesenchyme (*Jeong et al., 2004*), partly recapitulated the mandibular patterning defects in $Smo^{c/c}$;*Wnt1-Cre* and $Smo^{c/c}$;*Hand2-Cre* embryos, including tongue agenesis and ectopic ossification in the oral side of the mandibular mesenchyme. The $Foxf1^{c/c}$;$Foxf2^{c/c}$;*Wnt1-Cre* embryos also showed orally expanded activation of pSmad1/5/9 in the distal mandibular mesenchyme, indicating that these transcription factors act downstream of Shh signaling to pattern the oral-aboral axis through antagonizing BMP signaling. It has been shown that $Foxf2^{-/-}$ mutant embryos had increased pSmad1/5 and decreased pSmad2/3 in the developing brain tissues (*Reyahi et al., 2015*). Another report showed decreased

pSmad2/3 in the developing secondary palate in *Foxf2*⁻/⁻ mouse embryos (*Nik et al., 2016*). However, how Foxf1/2 regulates Smad activity is not known. None of these *Fox* family transcription factors have been shown to mediate Shh signaling function in anteroposterior patterning of the limb or dorsoventral patterning of the neural tube or somite. Thus, whereas antagonistic interactions of Shh and BMP signaling are crucial in patterning multiple tissues, the exact molecular mechanisms mediating hedgehog signaling induced antagonism of BMP signaling are context-dependent and the detailed molecular mechanism mediating hedgehog signaling function in oral-aboral patterning and its antagonism of BMP signaling in the developing mandibular arch requires further investigation.

Previous studies have shown that BMP signaling is crucial for proximal-distal patterning of the mandibular arch (*Liu et al., 2005*). In particular, Bmp4 acts in a dose-dependent manner to restrict *Fgf8* expression to the proximal mandibular arch epithelium (*Liu et al., 2005*). In addition, BMP signaling is required for activation of expression of *Hand1* in the distal cap mesenchyme of the mandibular arch through interacting with the Hand2 transcription factor, whose expression in the distal mandibular mesenchyme is activated by the Dlx5/6 transcription factors in response to endothelin-1 signaling (*Charité et al., 2001*; *Depew et al., 2002*; *Vincentz et al., 2016*). It has been shown that Hand2-mediated feedback down-regulation of Dlx5/6 in the distal mandibular mesenchyme is essential for both BMP-mediated activation of *Hand1* expression and for initiation of tongue morphogenesis (*Barron et al., 2011*; *Vincentz et al., 2016*). In addition, multiple signaling pathways, including TGF-beta, FGF, and Wnt, are involved in the interactions between the mandibular neural crest mesenchyme and tongue muscle precursor cells to regulate migration, proliferation, and differentiation of the myoblasts to organize into specific tongue muscles (*Parada and Chai, 2015*). We show that the tongue agenesis in the *Smo^{c/c}*;*Hand2-Cre* mutant embryos was preceded by reduced accumulation of *Myf5*-expressing myoblasts in the mandibular arch at E10.5, suggesting that hedgehog signaling in the neural crest cells is crucial for regulation of myoblast migration into the mandibular arch during tongue initiation. On the other hand, *Myf5*-expressing tongue precursor cells migrated normally to the mandibular arch in *Foxf1^{c/c}*;*Foxf2^{c/c}*;*Wnt1-Cre* embryos, but these mutant embryos still had severe disruption of tongue formation, suggesting that the Shh-Fox pathway acts in the mandibular neural crest mesenchyme to regulate multiple steps of tongue morphogenesis. How Shh-Foxf1/2 mediated antagonism of BMP signaling along the oral-aboral axis interacts with the BMP-Hand1/2-Dlx5/6 pathways in the distal mandibular mesenchyme to specify the distinct domains for proper tongue and dentary bone formation will require further investigation.

Although this study has not directly addressed which hedgehog ligand is required for patterning the oral-aboral axis of the mandibular arch, all available evidences indicate that Shh is the patterning signal in this process. Of the three mouse genes encoding hedgehog family ligands, only *Shh* has been shown expressed in the developing mouse mandibular arch prior to E10.5 (*Billmyre and Klingensmith, 2015*; *Jeong et al., 2004*). The tongue agenesis and ectopic ossification at the oral side of the mandibular mesenchyme in the *Smo^{c/c}*;*Wnt1-Cre* and *Smo^{c/c}*;*Hand2-Cre* embryos correlate well with the restricted expression of *Shh* in the oropharyngeal epithelium. Furthermore, it has been shown that tissue-specific inactivation of *Shh* in the oropharyngeal epithelium in mouse embryos led to tongue agenesis, distal truncation and severe hypoplasia of the mandible, which mimic the mandibular defects in the *Smo^{n/c}*;*Wnt1-Cre* and *Smo^{c/c}*;*Hand2-Cre* mice (*Billmyre and Klingensmith, 2015*; *Jeong et al., 2004*). Together, these data indicate that Shh ligands produced in the oropharyngeal epithelium signal through Smo in the mandibular mesenchyme to antagonize BMP signaling and pattern the oral-aboral axis of the mandibular arch.

Previous studies have indicated that Shh signaling plays a crucial role in the survival of neural crest-derived craniofacial mesenchyme (*Ahlgren and Bronner-Fraser, 1999*; *Billmyre and Klingensmith, 2015*; *Jeong et al., 2004*). Mouse embryos with either neural crest-specific deletion of *Smo* or oropharyngeal epithelium-specific deletion of *Shh* caused dramatically increased apoptosis of the mandibular arch mesenchyme cells (*Billmyre and Klingensmith, 2015*; *Jeong et al., 2004*). Billmyre and Klingensmith showed that pharmacological inhibition of p53 reduced apoptosis of the mandibular arch mesenchyme caused by loss of Shh signaling (*Billmyre and Klingensmith, 2015*). However, how loss of function of Shh-Smo signaling triggers p53-mediated apoptosis in the mandibular arch mesenchyme is unknown. In this study, we found that the significantly increased apoptosis of the distal mandibular arch mesenchyme in the *Smo^{c/c}*;*Hand2-Cre* embryos correlated with the increased BMP signaling, and that reduction in *Bmp4* gene dosage significantly reduced apoptosis of the distal mandibular arch mesenchyme in *Smo^{c/c}*;*Hand2-Cre*;*Bmp4^{+/-}* embryos. These results indicate that

Shh regulates neural crest survival in the mandibular arch at least in part through antagonizing BMP signaling. *Hayano et al., 2015* showed that ectopic activation of BMPR1A signaling in cranial neural crest cells caused a significant increase in p53 protein levels and induced p53-mediated apoptosis (*Hayano et al., 2015*). Taken together, these data indicate that Shh-Smo signaling regulates mandibular mesenchyme survival through antagonizing BMP signaling and preventing BMP-induced p53-mediated apoptosis.

In addition to crucial roles in cell survival and growth of the mandibular mesenchyme, Shh has previously been implicated in regulating development of Meckel's cartilage (*Billmyre and Klingensmith, 2015*; *Brito et al., 2008*). Tissue-specific deletion of *Shh* in the oropharyngeal epithelium caused complete lack of Meckel's cartilage formation in the mutant mouse embryos (*Billmyre and Klingensmith, 2015*). Conversely, ectopic expression of Shh in the mandibular arch ectoderm of early chick embryos resulted in formation of ectopic cartilage branching off from the endogenous Meckel's cartilage (*Haworth et al., 2007*). Moreover, grafting of Shh-producing quail embryonic fibroblasts in the presumptive proximal mandibular arch in chick embryos induced formation of two supernumerary Meckel's cartilages that develop in a mirror-image to the endogenous one (*Brito et al., 2008*). However, whereas both $Smo^{c/c}$;*Wnt1-Cre* and $Smo^{c/c}$;*Hand2-Cre* embryos exhibit distal truncation of the mandibular skeleton, Meckel's cartilage formed in these mutant embryos. *Jeong et al., 2004* showed that the $Smo^{c/n}$;*Wnt1-Cre* embryos also developed Meckel's cartilage, albeit distally truncated. Moreover, we found that the $Smo^{c/c}$;*Hand2-Cre*;$Bmp4^{+/-}$ embryos developed Meckel's cartilage and the cartilage symphysis in the distal region of the mandible. These results indicate that Shh either directly regulate Meckel's cartilage formation through a Smo-independent mechanism or indirectly regulate Meckel's cartilage formation through signaling in another cell type in the mandibular arch. It has been shown that exogenous Fgf8 was able to rescue Meckel's cartilage formation in embryonic mouse mandibular explant cultures treated with the hedgehog antagonist cyclopamine (*Melnick et al., 2005*). In mouse embryos with oropharyngeal epithelial deletion of *Shh*, *Fgf8* expression in the proximal mandibular arch epithelium was significantly reduced in comparison with control littermates (*Billmyre and Klingensmith, 2015*). Induction of supernumerary Meckel's cartilage formation in the chick mandibular arch by an extra source of Shh in the presumptive mandibular arch ectoderm was preceded by induction of ectopic expression of *Fgf8* and *Bmp4* in the caudal side of the mandibular arch ectoderm (*Brito et al., 2008*; *Haworth et al., 2007*). Taken together, these data indicate that Shh signaling regulates mandibular growth and patterning through at least two distinct mechanisms: (1) Shh signals to adjacent mandibular arch ectodermal cells to regulate expression of Fgf8 to control mandibular growth and Meckel's cartilage development, and (2) Shh signals directly through Smo in the neural crest-derived mandibular mesenchyme cells to regulate their survival and oral-aboral patterning through antagonizing BMP signaling.

## Materials and methods

### Mouse strains

The $Smo^{c/c}$, *R26SmoM2*, $Foxf1^{c/c}$, $Foxf2^{c/c}$, *Wnt1-Cre*, and *Hand2-Cre* mice have been described previously (*Bolte et al., 2015*; *Danielian et al., 1998*; *Jeong et al., 2004*; *Ren et al., 2014*; *Ruest et al., 2003*). The *Hand2-Cre* mice were maintained by crossing with *C57BL/6J* mice. *Wnt1-Cre* mice were maintained by crossing with CD1 (Charles river) mice. $Smo^{c/c}$, *R26SmoM2*, $Foxf1^{c/c}$ and $Foxf2^{c/c}$ mice were maintained by intercrossing homozygotes. Noon of the day a vaginal plug was identified was designated as embryonic day (E) 0.5. Only littermate control embryos were used in each experiment. No masking was used during group allocation, data collection and/or data analysis. No data was excluded in this study. This study was performed in strict accordance with the recommendations in the Guide for the Care and Use of Laboratory Animals by the National Institutes of Health. The animal use protocol was approved by the Institutional Animal Care and Use Committee of Cincinnati Children's Hospital Medical Center (Permit Number IACUC2016-0095).

### Preparation of single cell suspension

A psychrophilic protease protocol for single cell dissociation was modified from previous report (*Adam et al., 2017*). Mandibles from E10.5 wildtype CD1 mouse embryos were rapidly dissected in

ice-cold PBS. Five embryonic mandibles were pooled in a sterile 1.5 ml microcentrifuge tube with 350 µl of freshly made protease solution containing 5 mg/ml of *Bacillus Licheniformis* protease (Sigma P5380) in DPBS, 5 mM CaCl2, and 125 U/ml DNase (Applichem A3778). The sample was incubated in 4°C for a total 8 min, with trituration for 15 s every 2 min using a 1 ml pipet. Single cell dissociation was confirmed by observation of samples under microscope. Following inactivation of the protease with ice-cold DMEM containing 10% FBS, the cells were pelleted by centrifugation at 1200 g for 5 min. The cells were resuspended in 1 ml ice-cold DMEM containing 10% FBS, and filtered through a sterile 40 µM filter (BD Falcon 352340) and pelleted by centrifugation at 1200 g for 5 min. The dissociated cells were washed with ice-cold DMEM containing 10% FBS. Cell number and viability were analyzed using a hemocytometer following trypan blue (Gibco 15250061) staining. The single cell preparation displayed greater than 90% viability in this experiment.

## Single-cell RNA-seq: barcoding, cDNA amplification, library construction, and data matrix generation

The single cell suspension was adjusted to 900 cells/µl and approximately 14,000 cells were loaded onto a well on a 10x Chromium Single Cell instrument (10x Genomics). Barcoding, cDNA amplification and library construction were performed using the Chromium Single Cell 3′ Reagent Kits v2 exactly according to the manufacturer's instructions. This system uses the 10x GemCode technology that partitions thousands of cells individually into nanoliter-sized Gel Bead-In-Emulsions (GEMs), where cell lysis and cDNA synthesis occur, with all cDNAs generated from an individual cell sharing a common cell specific barcode. Unique Molecular Identifier (UMI) barcodes, specific for each oligonucleotide on a bead, allow compression of data to hybridization events, reducing amplification bias. Post cDNA amplification reaction and cleanup were performed using SPRI select reagent (Beckman Coulter, Cat# B23318). Post cDNA amplification and post library construction quality analyses were performed using the Agilent Bioanalyzer High Sensitivity kit (Agilent 5067–4626). The final single cell 3′ library contains standard Illumina paired-end constructs (begin and end with P5 and P7 primer sequences, and 16 bp 10x Barcode, 10 bp UMI, Read one primer sequence, Read two primer sequence, and the 8 bp i7 sample index.

Libraries were sequenced using an Illumina HiSeq 2500 and the paired-end 75 bp sequencing flow cell. Sequencing parameters used were: Read 1, 27 cycles; i7 index, eight cycles; Read 2, 147 cycles, according to manufacturer's recommendations, which produced about 300 million reads. The sequencing output raw data were processed using CellRanger 2.0.0 (http://10xgenomics.com) to obtain a gene-cell data matrix.

## Single-cell RNA-seq data processing, tSNE visualization, and cell clustering analysis

Once the gene-cell data matrix was generated, poor quality cells were excluded through cell filtering by selecting cells that expressed >1000 unique genes, with each gene expressed in a minimum of three cells. Only one read per cell was needed for a gene to be counted as expressed. Cells containing high percentages of mitochondrial (>10%), histone (>0.002%), or hemoglobin (>0.01%) genes were excluded. The resulting gene expression matrix was normalized to 10,000 molecules per cell and log transformed according to Macosko et al. (*Macosko et al., 2015*). The top 593 genes with the highest variability among cells were found using Seurat's FindVariableGenes function, x.low.cutoff = 0.0125, x.high.cutoff = 4, y.cutoff = 0.5. These highly variable genes were used for principal components (PCs) analysis using the Seurat package (version 2.0.1, R 3.4.1 library). The influence of the number of unique molecular identifiers was minimized using Seurat's RegressOut function. Clustering was performed with Seurat's t-distributed stochastic neighbor embedding (tSNE) implementation using the first 18 significant PCs determined by JackStraw plot (*Macosko et al., 2015*). All clustering was unsupervised, without driver genes. Cell groupings were determined by shared nearest neighbor (SNN) using the Louvain algorithm. The tSNE maps visualized the single cells on two-dimensional or three-dimensional space based on expression signatures of the variable genes. Marker genes were determined for each cluster using Seurat's FindAllMarkers function using the Wilcoxon rank sum test, only considering genes expressed in a minimum of 25% of cells and fold change threshold of 1.3. Over/under clustering was verified via gene expression heatmaps. Monocle-2 (*Qiu et al., 2017*) was used for analysis of developmental trajectories of the neural crest cell

population. The raw UMI gene counts were used and modeled on the negative binomial distribution using genes that were expressed in a minimum of 10 cells. The BEAM function was used to determine genes enriched in each trajectory state.

The scRNA-seq data from this study have been deposited into the National Center for Biotechnology Information Gene Expression Omnibus (NCBI GEO) database (accession number GSE112837).

## Histology, immunofluorescent staining, and quantification of apoptotic cells

For histological analysis, embryos were dissected at desired stages from timed pregnant mice, fixed in 4% paraformaldehyde (PFA), dehydrated through an ethanol series, embedded in paraffin, sectioned at 7 µm thickness, and stained with alcian blue followed by hematoxylin and eosin.

Detection of alkaline phosphatase was performed on frozen sections. Briefly, the sections were washed with NTMT (0.1 M NaCl, 0.1 M Tris-HCl pH 9.5, 50 mM MgCl2, 0.1% Tween20) buffer, then stained with NBT/BCIP Substrate Solution at 37°C.

Immunofluorescent staining of paraffin sections was performed following standard protocols. Three embryos of each genotype have been used for the experiment. Antibodies used were: rabbit Anti-Runx2 (Santa Cruz, sc-10758), Goat anti-Foxf1 (R&D Systems, AF4798), and sheep anti-Foxf2 (R&D Systems, AF6988). Anti muscle actin antibody (Clone HUC1-1) (*Sawtell and Lessard, 1989*) was provided by Dr. James Lessard (Cincinnati Children's Hospital Medical Center).

Immunofluorescent staining of pSmad1/5/9 (Cell Signaling, #13820) was performed on frozen sections following standard protocol with modifications. The number of pSmad1/5/9 positive nuclei over the total DAPI positive nuclei was recorded as percentage of pSmad1/5/9 positive cells using Nikon NIS-Elements AR software (Nikon). The mandibular mesenchyme was divided into three domains along the oral-aboral axis, and percentage of the pSmad1/5/9 positive cells was quantified in each domain, and for each genotype. The results were presented as mean ± SD. Statistical analysis was performed using two-way ANOVA followed by Bonferroni post-test to compare data from all domains. For *SmoM2;Hand2-Cre* mutants and control samples, the percentage of the pSmad1/5/9 positive cells throughout the oral-aboral axis of the distal mandibular mesenchyme on sagittal sections were quantified and statistical analysis was performed using Student's t-test. P value less than 0.05 was considered significant.

The quantification of cleaved Caspase3 signal was perform in $Smo^{c/c}$, $Smo^{c/c}$;Hand2-Cre and $Smo^{c/c}$;Hand2-Cre;Bmp4$^{+/-}$ embryos. Whole mount immunofluorescent staining of E10.5 mandible tissues was performed as previously described using rabbit anti-cleaved Caspase3 (BD Pharmingen, 559565) (*Xu et al., 2016a*). Immunostained samples were scanned by confocal microscopy using a Nikon A1 Inverted Microscope. Four embryos of each genotype, and 10–20 sections from each embryo were used for quantification. In total, 69 sections from $Smo^{c/c}$, 54 sections from $Smo^{c/c}$;Hand2-Cre, and 69 sections from $Smo^{c/c}$;Hand2-Cre;Bmp4$^{+/-}$ embryos were used for quantification. The results were presented as mean ± SD. Statistical analysis was performed with one-way ANOVA followed by Tukey post hoc test to compare all pairs. P value less than 0.05 was considered significant.

## Detection of β-Galactosidase activity, skeletal preparations and in situ hybridization assays

X-Gal staining of whole mount embryos and cryostat sections was performed as previously described (*Hogan, 1994*). Sections were counterstained with eosin. Skeletal preparations were performed as previously described (*Martin et al., 1995*). Whole mount and section in situ hybridization was performed as previously described (*Zhang et al., 1999*). At least three embryos of each genotype were hybridized to each probe and only probes that detected consistent patterns of expression in all samples were considered as valid results.

## Acknowledgements

We thank Dr. James Lessard at Cincinnati Children's Hospital Medical Center for generously providing the anti muscle antibody (Clone HUC1-1). This work was supported by National Institutes of

Health/National Institute of Dental and Craniofacial Research grant DE027046 and by Shriners Hospitals for Children grant #85900 to RJ.

## Additional information

### Funding

| Funder | Grant reference number | Author |
|---|---|---|
| National Institutes of Health | R01 DE027046 | Rulang Jiang |
| Shriners Hospitals for Children | #85900 | Rulang Jiang |

The funders had no role in study design, data collection and interpretation, or the decision to submit the work for publication.

### Author contributions

Jingyue Xu, Conceptualization, Resources, Data curation, Formal analysis, Validation, Investigation, Visualization, Methodology, Writing—original draft, Writing—review and editing, Conceived and designed the project, Performed experiments, Performed data analysis; Han Liu, Data curation, Formal analysis, Validation, Investigation, Visualization, Methodology, Writing—review and editing, Conceived and designed the project, Performed experiments, Performed data analysis; Yu Lan, Conceptualization, Data curation, Formal analysis, Funding acquisition, Investigation, Visualization, Methodology, Project administration, Writing—review and editing, Performed experiments, Performed data analysis; Mike Adam, Data curation, Software, Formal analysis, Visualization, Methodology, Writing—review and editing, Participated in design of scRNA-seq experiments, Provided protocol for single cell preparation, Performed data analysis; David E Clouthier, Conceptualization, Resources, Formal analysis, Supervision, Funding acquisition, Investigation, Methodology, Writing—review and editing, Provided the Hand2-Cre mouse strain, Participated in experimental design and data analysis; Steven Potter, Software, Formal analysis, Supervision, Investigation, Methodology, Writing—review and editing, Participated in design of scRNA-seq experiments, Provided protocol for single cell preparation, Performed data analysis; Rulang Jiang, Conceptualization, Resources, Formal analysis, Supervision, Funding acquisition, Validation, Investigation, Visualization, Methodology, Writing—original draft, Project administration, Writing—review and editing, Conceived and designed the project, Performed data analysis

### Author ORCIDs

Jingyue Xu (iD) http://orcid.org/0000-0002-1765-1884
Rulang Jiang (iD) http://orcid.org/0000-0001-7842-4696

### Ethics

Animal experimentation: This study was performed in strict accordance with the recommendations in the Guide for the Care and Use of Laboratory Animals by the National Institutes of Health. The animal use protocol was approved by the Institutional Animal Care and Use Committee of Cincinnati Children's Hospital Medical Center (Permit Number IACUC2016-0095).

### Decision letter and Author response

Decision letter https://doi.org/10.7554/eLife.40315.037
Author response https://doi.org/10.7554/eLife.40315.038

## Additional files

### Supplementary files

• Supplementary file 1. List of marker genes with more than 1.5-fold enrichment in the individual clusters. Column A lists gene name. Column B lists p value of differential expression. Column C lists average fold change over all other clusters. Column D list the percentage of cells in the

corresponding cluster expressing the marker gene. Column E list the percentage of cells in all other clusters combined expressing the marker gene. Column F list the Bonferroni corrected p value. Column G lists the cluster name. NC1 – NC2, neural crest derived mesenchyme cells; Epi, epithelial cells; Endo, endothelial cells; HM, head mesoderm cells.

DOI: https://doi.org/10.7554/eLife.40315.026

• Supplementary file 2. List of marker genes exhibiting differential expression (at least 1.3-fold) between cells in the NC3 cluster and cells in the NC1 and NC2 clusters. Column A list gene name. Column B list p value of differential expression. Column C lists average fold change of expression of the marker gene in NC1/2 cells over NC3 cells. Positive value in Column C indicates higher levels of expression in NC1/2 than in NC3. Column D lists percentage of cells in NC1/2 clusters expressing the gene. Column E list percentage of cells in NC3 cluster expressing the gene. Column F list Bonferroni corrected p value of differentiation expression. Genes whose expression pattern is shown in *Figure 1—figure supplement 4* are highlighted in yellow.

DOI: https://doi.org/10.7554/eLife.40315.027

• Supplementary file 3. List of marker genes exhibiting more than 1.3-fold enrichment in expression levels in a specific neural crest subgroup over all other five subgroups. Genes that are shown in *Figure 1B* are highlighted in yellow color. Column A lists gene name. Column B lists p value of differential expression. Column C lists average fold change over all other subgroups. Column D list the percentage of cells in the corresponding subgroup expressing the marker gene. Column E list the percentage of cells in all other subgroups combined expressing the marker gene. Column F list the Bonferroni corrected p value of differential expression. Column G lists the subgroup number corresponding to *Figure 1B*.

DOI: https://doi.org/10.7554/eLife.40315.028

• Supplementary file 4. Top 50 hits from gene ontology (GO) analyses of marker genes of Subgroup 0 of the neural crest cells shown in *Figure 1B*.

DOI: https://doi.org/10.7554/eLife.40315.029

• Supplementary file 5. Top 100 hits from gene ontology (GO) analyses of marker genes of Subgroup 1 of neural crest cells shown in *Figure 1B*. GO analysis was performed using Toppgene (https://toppgene.cchmc.org/enrichment.jsp).

DOI: https://doi.org/10.7554/eLife.40315.030

• Supplementary file 6. Top 50 hits from gene ontology (GO) analyses of marker genes of State three from developmental trajectory analysis shown in *Figure 1—figure supplement 7*.

DOI: https://doi.org/10.7554/eLife.40315.031

• Supplementary file 7. Top 20 hits from gene ontology (GO) analyses of marker genes of State four from developmental trajectory analysis shown in *Figure 1—figure supplement 7*.

DOI: https://doi.org/10.7554/eLife.40315.032

• Transparent reporting form

DOI: https://doi.org/10.7554/eLife.40315.033

## Data availability

The single-cell RNA-seq data from this study have been deposited into the National Center for Biotechnology Information Gene Expression Omnibus (NCBI GEO) database (accession number GSE112837). All data generated or analyzed during this study are included in the manuscript and supporting files.

The following dataset was generated:

| Author(s) | Year | Dataset title | Dataset URL | Database and Identifier |
|---|---|---|---|---|
| Xu J | 2018 | Hedgehog signaling patterns the oral-aboral axis of the mandibular arch | https://www.ncbi.nlm.nih.gov/geo/query/acc.cgi?acc=GSE112837 | NCBI Gene Expression Omnibus, GSE112837 |

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
