## [Decision Letter]

Thank you for submitting your article "Sonic hedgehog signaling patterns the oral-aboral axis of the mandibular arch" for consideration by *eLife*. Your article has been reviewed by four peer reviewers, including Shahragim Tajbakhsh as guest Reviewing Editor, and the evaluation has been overseen by Didier Stainier as the Senior Editor. The following individuals involved in review of your submission have agreed to reveal their identity: Ramkumar Sambasivan (Reviewer #1).

The reviewers have discussed the reviews with one another and the Reviewing Editor has drafted this decision to help you prepare a revised submission.

Summary:

This study reports on the role of Hedgehog signalling in the patterning of the oral-aboral axis in the mouse developing jaw. The authors use scRNAseq and identify genes in distinct cell populations in the mandibular arch and map these subsets in the embryo using in situ hybridisation. Using several genetically modified mouse models, the authors propose that hedgehog signaling antagonizes Bmp signalling in the presumptive osteogenic domain, which is critical for cell survival in the distal mandibular arch. The work complements the current understanding of FGF8 / BMP4 antagonism in patterning the proximo-distal axis of jaw, the experiments are well-designed and the extensive phenotypic data represents a significant advance in the field.

Essential revisions:

1) The presentation of scRNAseq data is not well integrated and the analysis appears preliminary. Additional analysis would enable the authors to support their conclusions, and would make the article more accessible to a wider readership.

2) Are the subdivisions of the mandibular arch sharply defined or extremes of expression gradients? In the in situ hybridizations, many genes show graded expression, and the authors also mention the presence of nested spatial expression patterns (subsection “Single-cell RNA-seq analysis of the E10.5 mouse mandibular arch”, third paragraph). It is not clear why the authors choose to represent the complex 3D organization of the mandibular arch with a simple 2D tSNE projection, and why they look for distinct clusters instead of analyzing expression gradients. Indeed, the genes shown in Figure 2 do match the four clusters shown in Figure 1B only in some cases. Some of the genes demarcate parts of these clusters (e.g. *pou3f3, foxf2*) or span subparts of multiple clusters (e.g. *lhx8*, covering part of cluster 0 and 3). Moreover, it is not evident why 3 stands as its own cluster: the heatmap in Figure 1C does not show genes specific of this cluster.

The dataset seems rich enough to characterise and display more clearly the three major spatial axes of the developing mandibular arch: oral-aboral, proximo-distal and rostro-caudal. The authors could explore computational methods better suited for this, for example PCA, ICA or diffusion maps. In this way the three axes might show up as separate components in PCA, ICA and/or diffusion map space. This would enable the authors to study the cells along "pseudospace" trajectories, and to plot more clearly the expression of genes with graded and nested expression along these axes. They should also identify in an unbiased way the gene families overrepresented along these axes (e.g. by GO-enrichment), and leverage their data further to support the main conclusions (e.g., are there genes involved in ossification vs. muscle differentiation activated at the two opposite extremes?).

3) The increased apoptosis found in the distal region of mandibular arch at E10.5 in Hand2Cre;Smoc/c may be responsible for the lack of distal mandibular skeletal structure at newborn stage. It would be helpful to discuss or speculate on the potential cellular mechanism that resulted in the ectopic dentary bone formation in Smo mutant mice.

4) The authors attribute the NC3 population from the single cell RNA-seq to a population of neural crest cells which underwent an incomplete lysis of their nucleus, thus clustering out of the NC1 and NC2 clusters. However it is considered that nuclear transcripts represent a small fraction of cellular transcripts, and that nuclear mRNA is a good substitute for cellular mRNA in scRNAseq cell type identification (Lake et al., Sci Rep, 2017). Could the lack of nuclear transcripts alone account for the clustering of this population ? Among the nuclear lncRNA, Sox11 seems to be highly differentially expressed in NC3 compared with NC1/2. Sox11 is known to be expressed in the palate 3 days later, as well as in nerves. Could this population come from a proximal contamination from a domain shared with the maxillary prominence or neurogenic cells in the mandible ?

5) The authors present a model of an oral endodermal Shh expression, leading to an activation of Foxf1/2 in the adjacent mesenchyme and inhibition of BMP4 signalling, controlling the expansion of the Msx1 domain.

- Importantly, do the authors observe an expansion of p-SMAD1/5/9 in the Wnt1-Cre; Foxf1/2 cKO?

- If so, can the authors speculate on the possible targets of Foxf1/2, leading to the inhibition of Smads?

6) The authors describe tongue agenesis in Wnt1-Cre;Smo cKO, Hand2-Cre; Smo cKO and Wnt1-Cre; Foxf1/2 cKO, but show only rostral frontal sections.

- Is tongue agenesis observed all along the oral rostro-caudal axis in the different cKO specimens?

- Do the authors have available data at E10.5 showing altered hypoglossal cord formation in cKO specimens?

- From the data collected in this study, the authors should discuss more extensively the role of the neural crest-derived population during tongue morphogenesis as previously proposed and reviewed in Parada and Chai (2015).

---

## [Author Response]

Essential revisions:1) The presentation of scRNAseq data is not well integrated and the analysis appears preliminary. Additional analysis would enable the authors to support their conclusions, and would make the article more accessible to a wider readership.

We thank the editors and reviewers for providing suggestions for re-analysis of our scRNA-seq data. For this manuscript revision, we have greatly expanded analysis of the scRNA-seq data using multiple methods and report the results in the first two figures in the manuscript body plus 8 supplementary figures, 7 supplementary tables, and a supplementary video for 3D tSNE projection. We re-wrote the Results section to report these new analyses and results. The results not only support the conclusions of the manuscript but also provide seamless and logical transition from the scRNA-seq analysis to our extensive mechanistic genetic studies in the rest of the manuscript.

2) Are the subdivisions of the mandibular arch sharply defined or extremes of expression gradients? In the in situ hybridizations, many genes show graded expression, and the authors also mention the presence of nested spatial expression patterns (subsection “Single-cell RNA-seq analysis of the E10.5 mouse mandibular arch”, third paragraph). It is not clear why the authors choose to represent the complex 3D organization of the mandibular arch with a simple 2D tSNE projection, and why they look for distinct clusters instead of analyzing expression gradients. Indeed, the genes shown in Figure 2 do match the four clusters shown in Figure 1B only in some cases. Some of the genes demarcate parts of these clusters (e.g. pou3f3, foxf2) or span subparts of multiple clusters (e.g. lhx8, covering part of cluster 0 and 3). Moreover, it is not evident why 3 stands as its own cluster: the heatmap in Figure 1C does not show genes specific of this cluster.The dataset seems rich enough to characterise and display more clearly the three major spatial axes of the developing mandibular arch: oral-aboral, proximo-distal and rostro-caudal. The authors could explore computational methods better suited for this, for example PCA, ICA or diffusion maps. In this way the three axes might show up as separate components in PCA, ICA and/or diffusion map space. This would enable the authors to study the cells along "pseudospace" trajectories, and to plot more clearly the expression of genes with graded and nested expression along these axes. They should also identify in an unbiased way the gene families overrepresented along these axes (e.g. by GO-enrichment), and leverage their data further to support the main conclusions (e.g., are there genes involved in ossification vs. muscle differentiation activated at the two opposite extremes?).

We have re-analyzed the scRNA-seq data and replaced Figure 1B and Figure 1C with new results. Instead of always reducing the tSNE dimensions to two, we have analyzed 3-dimensional tSNE projections. Although tSNE plots usually would show clusters of cell types arranged in a manner that is completely unrelated to their in vivo spatial relationships, our new iterative clustering analysis of the mandibular arch neural crest scRNA-seq data resulted in grouping the neural crest cells into six clusters that are distributed along three major spatial axes in a 3D tSNE projection, including the previously defined proximal-distal and rostral-caudal axes and the new experimentally validated oral-aboral axis (the new Figure 1B and Figure 1—video 1). This is probably because the extensive nested patterns and gradients of gene expression across the mandibular arch neural crest cell population provided abundant intermediate connecting cell types, resulting in a 3D tSNE projection that recapitulated the cellular distribution of the in vivo spatial organization. Thus, although the clusters do not have sharp boundaries and do not match most of the individual gene expression domains, they represented the patterns of cellular distribution along the major spatial axes at high resolution, with each cluster marked by multiple highly differentially expressed marker genes (new Figure 1C and Supplementary file 3). Unlike the original Figure 1B and Figure 1C, there is no discrepancy between the clustering tSNE projection in the new Figure 1B and the heatmap presented in Figure 1C. The new analysis results generated the main hypothesis regarding Hedgehog and BMP signaling in patterning the oral-aboral axis that was tested through extensive genetic studies reported in the rest of the manuscript. Thus, the expanded tSNE analysis, particularly with the use of 3D tSNE projection, significantly improved our decoding and interpretation of the scRNA-seq data and provide crucial support for the conclusions of the manuscript.

In the revised manuscript, we discuss the limitation of the tSNE clustering with mostly nonexclusive marker genes. We also revised Figure 2 to show direct comparisons of representative marker gene expression profiles in the tSNE plots with in vivo expression patterns detected by whole mount in situ hybridization. Moreover, we also performed principal component analysis as well as developmental trajectory analysis (using Monocle 2) of the scRNA-seq dataset, as suggested, but neither methods allowed clear identification of spatial patterns of the neural crest derived mandibular arch mesenchyme, most likely due to the fact that most of the E10.5 mouse mandibular mesenchyme cells are highly proliferating undifferentiated progenitor cells that have extensive overlapping transcriptome profiles. We present the PCA and trajectory analyses results in Figure 1—figure supplement 5-8, and additional GO analysis results in Supplementary files 4-7.

3) The increased apoptosis found in the distal region of mandibular arch at E10.5 in Hand2Cre;Smoc/c may be responsible for the lack of distal mandibular skeletal structure at newborn stage. It would be helpful to discuss or speculate on the potential cellular mechanism that resulted in the ectopic dentary bone formation in Smo mutant mice.

Our data presented in Figure 8 agree with the hypothesis that increased apoptosis in the distal region of the mandibular arch may be responsible for the lack of distal mandibular skeletal structures at the new born stage and showed that decreasing BMP4 gene dosage partly rescued distal mandibular skeletal structures. Together with the other data presented in this manuscript, we conclude that BMP signaling is crucial in patterning the oral-aboral axis as well as in hedgehog mediated mandibular mesenchymal survival. The original manuscript already included a paragraph discussing the mechanism involving BMP signaling in regulating mandibular mesenchyme survival. We regard the ectopic dentary bone formation as a secondary consequence of loss of the oral fate and disruption of oral-aboral patterning of the mandibular mesenchyme (Discussion, first paragraph).

4) The authors attribute the NC3 population from the single cell RNA-seq to a population of neural crest cells which underwent an incomplete lysis of their nucleus, thus clustering out of the NC1 and NC2 clusters. However it is considered that nuclear transcripts represent a small fraction of cellular transcripts, and that nuclear mRNA is a good substitute for cellular mRNA in scRNAseq cell type identification (Lake et al., Sci Rep, 2017). Could the lack of nuclear transcripts alone account for the clustering of this population ? Among the nuclear lncRNA, Sox11 seems to be highly differentially expressed in NC3 compared with NC1/2. Sox11 is known to be expressed in the palate 3 days later, as well as in nerves. Could this population come from a proximal contamination from a domain shared with the maxillary prominence or neurogenic cells in the mandible ?

We agree that the nuclear transcripts represent a fraction of cellular transcripts and nuclear mRNA is a good substitute for cellular mRNA in scRNA-seq cell type identification. We have expanded the analysis comparing the NC3 cluster with all other clusters and show that NC3 cluster cells showed significantly reduced overall transcripts and the number of genes expressed per cell than all other clusters (new Figure1—figure supplements 2 and 3). Furthermore, we show by tSNE plots that NC3 cluster cells had very low or absence of ubiquitously highly expressed nuclear lncRNAs (new Figure1—figure supplement 4). Since we do not know what percentage of the total cellular transcripts are nuclearly localized in the E10.5 mandibular mesenchyme cells, we do not know whether lack of nuclear transcripts alone could account for the clustering of the NC3 population. However, the low overall transcripts and number of expressed genes detected per cell, together with the unique lack of nuclear lncRNAs highly expressed in other cell clusters, strongly suggest that the NC3 cells were incompletely lysed and revised our interpretation to “these data suggest that the NC3 cluster resulted most likely from incomplete lysis of a subset of single cells”. We also show in Figure1—figure supplement 4 that Sox11 expression is obviously lower in NC3 cell than in the other clusters. Thus, it is unlikely for NC3 to be due to contamination from the maxillary prominence. To our knowledge, this finding of failure of nuclear lysis in a subset of cells in scRNA-seq is new and should be of use to the single cell analysis community.

5) The authors present a model of an oral endodermal Shh expression, leading to an activation of Foxf1/2 in the adjacent mesenchyme and inhibition of BMP4 signalling, controlling the expansion of the Msx1 domain.- Importantly, do the authors observe an expansion of p-SMAD1/5/9 in the Wnt1-Cre; Foxf1/2 cKO?- If so, can the authors speculate on the possible targets of Foxf1/2, leading to the inhibition of Smads?

We have performed new experimental analysis as requested and present the pSmad1/5/9 data in the Foxf1/2 cKO embryos in the new Figure 10. Indeed, our new data show that the Foxf1/2 cKO embryos had orally expanded activation of pSmad1/5/9 in the mandibular arch mesenchyme (new Figure 10). We also added a brief discussion of possible mechanisms involving Foxf1/2 mediated regulation of the Smads in the Discussion section (second paragraph).

6) The authors describe tongue agenesis in Wnt1-Cre;Smo cKO, Hand2-Cre; Smo cKO and Wnt1-Cre; Foxf1/2 cKO, but show only rostral frontal sections.- Is tongue agenesis observed all along the oral rostro-caudal axis in the different cKO specimens?- Do the authors have available data at E10.5 showing altered hypoglossal cord formation in cKO specimens?- From the data collected in this study, the authors should discuss more extensively the role of the neural crest-derived population during tongue morphogenesis as previously proposed and reviewed in Parada and Chai (2015).

We have analyzed the tongue structures using serial sagittal and frontal sections in the *Smo* cKO and *Foxf1/f2 cKO* mutant embryos. We used immunofluorescent detection of muscle actin to label the tongue muscles. The anterior two-thirds of the tongue are located in the oral cavity, which arise from the mandibular arch, whereas the posterior third, known as the pharyngeal part, is derived from the third branchial arches. We found that all the mutant samples exhibited severe tongue defects, with only rudimentary tongue tissue in the posterior pharyngeal part but lack the oral part of the tongue tissues (Figure4—figure supplement 1 and Figure 6—figure supplement 1). In the *Foxf1^c/c^;Foxf2^c/c^;Wnt1Cre* embryos, the remaining muscular structures extend to more anterior region, but we didn’t detect outgrowth or protrusion of the oral tongue structures (Figure 9—figure supplement 1).

It has been shown that Myf5-expressing tongue muscle precursor cells migrate from the occipital somites into the developing mandibular arch along the hypoglossal cord by E10.5 and significantly reduced number of Myf5-expressing muscle precursor cells accumulated at the midline of the mandibular arch in E10.5 *Smo^c/n^;Wnt1Cre* mutantembryos (Jeong et al., 2004). We performed *Myf5* whole mount *in situ* hybridization in *Smo^c/c^;Hand2Cre* and *Foxf1^c/c^;Foxf2^c/c^;Wnt1Cre* embryos and compared with their control littermates. Myf5-expressing tongue muscle precursor cells were dramatically decreased in the mandibular arch in E10.5 *Smo^c/c^;Hand2Cre* embryos (Figure 6—figure supplement 1A-B), similar to that reported for *Smo^c/n^;Wnt1Cre* mutantembryos. In the *Foxf1^c/c^;Foxf2^c/c^;Wnt1Cre* embryos, however, the Myf5-expressing tongue muscle precursor cells accumulated normally in the mandibular arch by E10.5 (Figure 9—figure supplement 1A-B) but formation of the oral tongue was still severely disrupted. These results further support that Foxf1/2 plays a major role in mediating hedgehog-dependent specification of the oral fate of the neural crest-derived mandibular arch mesenchyme. We have added a brief discussion about the Shh-Foxf1/2 pathway in patterning the mandibular neural crest cells and secondarily affecting tongue formation, citing Parada and Chai (2015) (Discussion, third paragraph).